# Trim41 is required to regulate chromosome axis protein dynamics and meiosis in male mice

Seiya Oura[1,2], Toshiaki Hino[3], Takashi Satoh[4,5,6], Taichi Noda[1,7,8], Takayuki Koyano[9], Ayako Isotani[1,10], Makoto Matsuyama[9], Shizuo Akira[5,6], Kei-ichiro Ishiguro[11], Masahito Ikawa[1,2,12,13]*

1 Research Institute for Microbial Diseases, Osaka University, Osaka, Japan, 2 Graduate School of Pharmaceutical Sciences, Osaka University, Osaka, Japan, 3 Department of Biological Sciences, Asahikawa Medical University, Asahikawa, Japan, 4 Department of Host Defense, Research Institute for Microbial Diseases, Osaka University, Osaka, Japan, 5 Laboratory of Host Defense, WPI Immunology Frontier Research Center, Osaka University, Osaka, Japan, 6 Department of Immune Regulation, Graduate School of Medical and Dental Sciences, Tokyo Medical and Dental University, Tokyo, Japan, 7 Priority Organization for Innovation and Excellence, Kumamoto University, Kumamoto, Japan, 8 Division of Reproductive Biology, Institute of Resource Development and Analysis, Kumamoto University, Kumamoto, Japan, 9 Division of Molecular Genetics, Shigei Medical Research Institute, Okayama, Japan, 10 Division of Biological Science, Graduate School of Science and Technology, Nara Institute of Science and Technology, Nara, Japan, 11 Department of Chromosome Biology, Institute of Molecular Embryology and Genetics (IMEG), Kumamoto University, Kumamoto, Japan, 12 The Institute of Medical Science, The University of Tokyo, Minato-ku, Tokyo, Japan, 13 Center for Infectious Disease Education and Research (CiDER), Osaka University, Osaka, Japan

* ikawa@biken.osaka-u.ac.jp

**Data Availability Statement:** All relevant data are within the manuscript and its Supporting Information files.

## Abstract

Meiosis is a hallmark event in germ cell development that accompanies sequential events executed by numerous molecules. Therefore, characterization of these factors is one of the best strategies to clarify the mechanism of meiosis. Here, we report tripartite motif-containing 41 (TRIM41), a ubiquitin ligase E3, as an essential factor for proper meiotic progression and fertility in male mice. Trim41 knockout (KO) spermatocytes exhibited synaptonemal complex protein 3 (SYCP3) overloading, especially on the X chromosome. Furthermore, mutant mice lacking the RING domain of TRIM41, required for the ubiquitin ligase E3 activity, phenocopied Trim41 KO mice. We then examined the behavior of mutant TRIM41 (ΔRING-TRIM41) and found that ΔRING-TRIM41 accumulated on the chromosome axes with overloaded SYCP3. This result suggested that TRIM41 exerts its function on the chromosome axes. Our study revealed that Trim41 is essential for preventing SYCP3 overloading, suggesting a TRIM41-mediated mechanism for regulating chromosome axis protein dynamics during male meiotic progression.

## Author summary

During the meiotic prophase, germ cells undergo various characteristic events such as a physical linkage of homologous chromosomes (termed synapsis). This linkage of

**Funding:** This work was supported by: the Japan Society for the Promotion of Science (JSPS) KAKENHI grants (JP19J21619 to S.O., JP19H05758 to T.H., JP18K14612 and JP20H03172 to T.N., 19H05743 to KI, and JP19H05750 and 21H04753 to M.I.); Takeda Science Foundation Grants to T.N. and M.I.; The Nakajima Foundation to T.N.; and the Bill & Melinda Gates Foundation (INV-001902 to M.I.). The funders had no role in study design, data collection and analysis, decision to publish, or preparation of the manuscript.

**Competing interests:** The authors declare that they have no conflict of interest.

homologous chromosomes is known to be established by a homology search. However, most mammals have heteromorphic sex chromosomes (X and Y) with a small region of homology, which entails exceptional and specialized responses such as physical sequestration into a membrane-less organelle (termed XY body). Characterization of essential molecules will be the key to clarifying the mechanism of these events and meiosis. In this study, we identified tripartite motif-containing 41 (TRIM41) a ubiquitin ligase E3, as an essential factor for proper synapsis configuration and XY body formation. The most striking phenotype of *Trim41*-deficient spermatocytes was a misregulation of axis protein dynamics such as synaptonemal complex protein 3 (SYCP3) and SYCP1, which was mainly observed on the X chromosome. These results suggested that TRIM41 might be essential for chromosome axis remodeling during meiotic progression.

## Introduction

Tripartite motif (TRIM) family proteins consist of more than 70 members in humans and mice [1–3] (also according to NCBI and MGI databases). They contain three zinc-binding domains in the N-terminus, a RING finger domain, one or two B-Box (B1/B2) motifs, and a coiled-coil (CC) region (Fig 1A). The RING domains of many TRIM family members have been shown to confer E3 ubiquitin ligase activity [3,4] (Fig 1B) and are implicated in various biological functions such as innate immunity and carcinogenesis regulation [5–7]. TRIM41-mediated ubiquitination also functions in natural immunity via target protein degradation, including B-cell leukemia/lymphoma 10 (BCL10) [8], and nucleoproteins of the influenza A virus and vesicular stomatitis virus [9,10]. Therefore, to examine the effects in natural immunity, we produced *Trim41* knockout (KO) mice. However, unexpectedly, *Trim41* KO male mice exhibited complete infertility due to meiotic defects. Thus, we focused on the analysis of meiotic events in this report.

Meiosis is a dynamic chromosome event in germ cell development. Upon entry into meiosis, in the leptotene stage, programmed DNA double-strand breaks (DSBs) are introduced [11], enabling a genome-wide search for homology through the repair process. This homologous search drives the pairing and alignment, finally leading to physical juxtaposition (called synapsis) of all homologs in the pachytene stage, although it should be noted that a significant amount of chromosome pairing occurs before the initiation of DSBs [12]. However, unlike autosomes that fully synapse between homologs, sex chromosomes (X and Y in mammals) only synapse at the pseudoautosomal region (PAR) and are sequestered into a physically separated nuclear territory known as the XY body [13]. The molecules involved in these complex events are not fully characterized yet, especially in mammals, due to difficulties in culturing and genetically manipulating spermatogenic cells *in vitro*. Thus, physiological screening with KO mice has been a powerful strategy to identify meiosis-related factors [14–16].

In this study, we revealed that *Trim41* deficiency cause overloading of synaptonemal complex protein 3 (SYCP3) on several chromosome axes, especially on the X chromosome. A *Trim41* transgene with germ cell-specific expression rescued the spermatogenesis defects, which showed that TRIM41 directly regulates meiosis. Further, deletion of the RING domain (ΔRING) phenocopied *Trim41* KO. More importantly, ΔRING-TRIM41 accumulated on chromosome axes with overloaded SYCP3, suggesting that TRIM41-mediated protein degradation may act in removing overloaded SYCP3. Thus, our study demonstrated that mammalian spermatocytes have a TRIM41-mediated mechanism for chromosome axes remodeling.

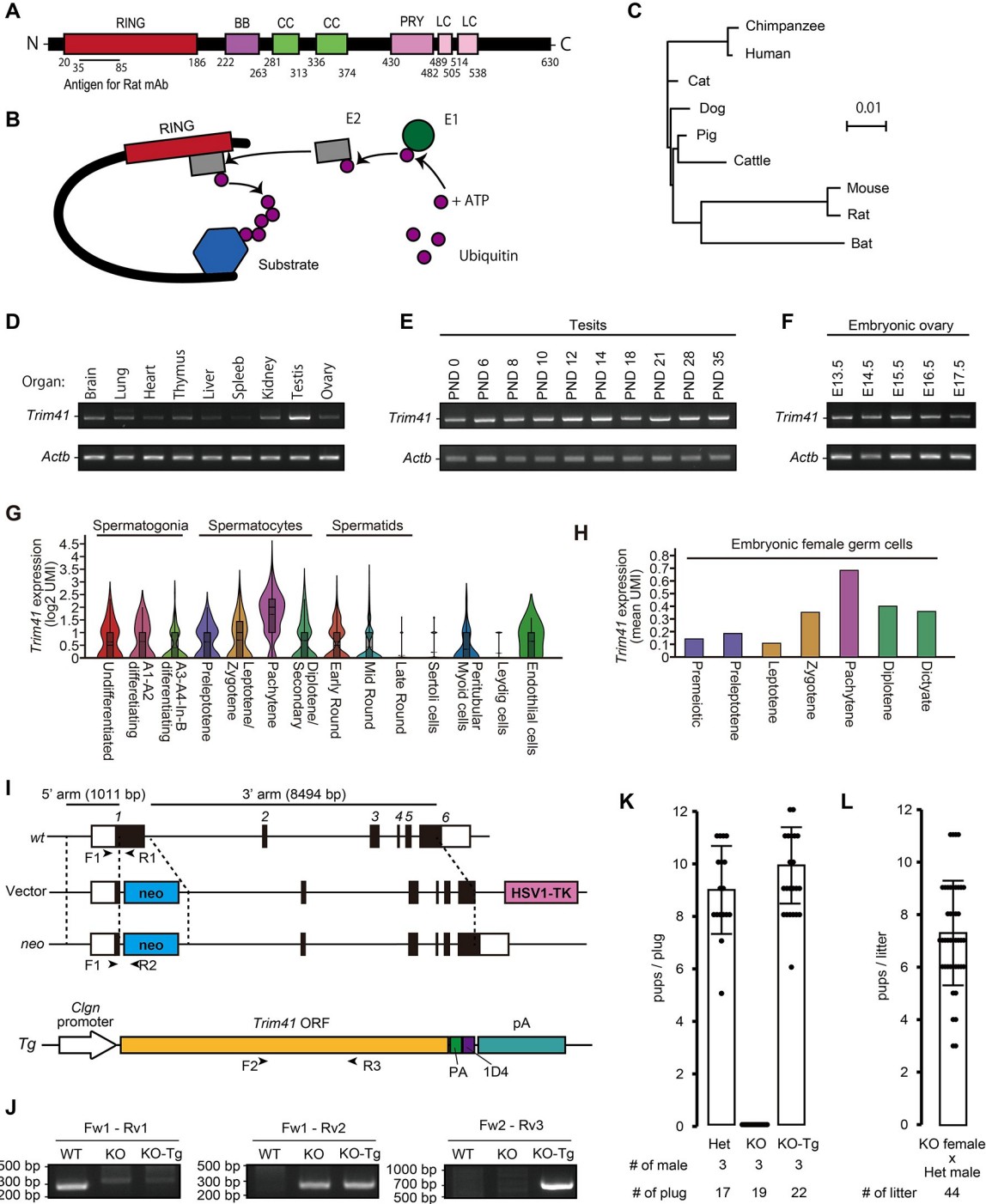

**Fig 1. Production of *Trim41* KO mice and fertility analysis.** (A) Schematic of TRIM41 protein structure and antigen position. (B) Model of TRIM E3 ubiquitin ligase activity. The RING domain at the N terminus interacts with E2 ubiquitin-conjugating enzyme. The C terminal portion of TRIM recognizes the substrates. (C) Phylogenetic tree constructed by ClustalW with TRIM41 sequences of various mammals. (D) RT-PCR using multi-tissue cDNA. *Actb* was used as a loading control. (E) RT-PCR using postnatal testis cDNA. *Actb* was used as a loading control. (F) RT-PCR using embryonic ovary cDNA. *Actb* was used as a loading control. (G) The *Trim41* expression profile between testicular cells based on published single-cell RNA sequencing data, visualized by 10 x genomics Loupe Browser. UMI means Unique Molecular Identifiers. (H) The *Trim41* expression profile between embryonic female germ cells based on published single-cell RNA sequencing data. (I) Targeting scheme for *Trim41* disruption and a Tg construct. Black and white boxes in the gene map indicate coding and non-coding regions, respectively. Black arrowheads (Primer F1, F2, R1, R2, and R3) indicate primers used for genotyping. (J) An example of genotyping PCR with primer sets in G. (K) The result of mating tests. Pups/plug: 8.9±1.7 [Het]; 0 [KO];

9.4±1.6 [KO-Tg] (s.d.). Error bars indicate standard deviation. The numerical data are available in S3 Table (L) Pup numbers from mating pairs of *Trim41* KO females and *Trim41* Het males (7.4±2.3; s.d.). Error bars indicate standard deviation. The numerical data are available in S3 Table.

## Results

### *Trim41* is evolutionarily conserved between mammals and highly expressed in pachytene germ cells

The Treefam database (http://www.treefam.org/; Release 9, March 2013) shows that TF342569, a family containing TRIM41, has been annotated in 50% of eukaryotes, 98% of vertebrates, and 100% of mammals (S1A Fig). Also, phylogenetic analysis with Clustal W2.1 [17] showed that TRIM41 was highly conserved in many mammals, including cattle, dogs, mice, and humans (Figs 1C and S1B). Next, to determine the tissue expression profile of *Trim41*, we performed RT-PCR using cDNA obtained from adult tissues. These RT-PCR showed that *Trim41* is expressed ubiquitously, albeit most highly in testis (Fig 1D). Then to examine the expression timing in germ cells, we performed RT-PCR using postnatal testis and embryonic ovary and detected the strongest PCR signal around postnatal day (PND) 14 of testis and embryonic day (E) 15.5 of ovary (Fig 1E and 1F), both of which correspond the pachytene stage. Furthermore, according to published single-cell RNA-sequencing (scRNA-seq) data analyzing mouse testis [18] and embryonic ovary [19], *Trim41* was most highly expressed in pachytene germ cells (Fig 1G and 1H). These results suggest that TRIM41 functions during the meiotic phase of mammalian gametogenesis.

### *Trim41* is essential for male fertility

To uncover the function of *Trim41* in vivo, we generated *Trim41* KO mice by transfecting embryonic stem cells (ESCs) with a targeting vector (Fig 1I; vector). Targeted ESC clones were injected into ICR embryos to obtain chimeric mice. The chimeric mice were mated with wild-type (WT) females to establish *Trim41* KO lines (Fig 1J). KO mice obtained by heterozygous intercrosses showed no overt gross defects in development, behavior, and survival. We housed individual *Trim41* KO male mice with WT females for two months to analyze their fertility. Although we observed 19 vaginal plugs, *Trim41* KO males failed to sire pups (Fig 1K). On the other hand, we obtained pups from the mating of *Trim41* KO females with *Trim41* heterozygous (Het) males (7.4±2.3; Fig 1L), indicating that *Trim41* is dispensable for female fertility. As *Trim41* Het male mice are fully fertile, we used littermate heterozygous males as controls in some experiments.

### A *Trim41* transgene under *Clgn* promoter restored infertility in *Trim41* KO males

To rule out the possibility that an aberrant genetic modification near the *Trim41* locus and/or latent systemic abnormalities affected male fertility, we carried out a tissue-specific rescue experiment by generating transgenic (Tg) mouse lines. First, we injected a DNA construct having PA-1D4-tagged *Trim41* under the testis-specific *Clgn* promoter [20] (Fig 1G; Tg) and established a Tg line (Fig 1H; Fw2-Rv3). *Clgn* promoter-driven *Trim41* expression began around postnatal day (PND) 9–11 (S2A Fig), corresponding to spermatocyte appearance. The Tg expression was also confirmed by immunoblotting analysis using anti-PA (S2B Fig) and -TRIM41 antibody (S2C Fig), although the expression level of the Tg was lower than that of intrinsic *Trim41* (S2C Fig; WT v.s. KO-Tg). We housed *Trim41* KO male mice expressing the Tg (referred to as KO-Tg) with WT females and observed pups produced at comparable

number to control (Fig 1I), showing the *Clgn* promoter-driven *Trim41* Tg rescued the fertility of KO males. These results indicated that *Trim41* expression from meiotic entry onward in testis is essential for male fertility.

## *Trim41* KO male mice exhibited oligozoospermia

When we observed gross testis morphology, *Trim41* KO testes were smaller than those of control (testis/ body weight: $3.2\pm0.3 \times 10^{-3}$ [WT], $3.2\pm0.2 \times 10^{-3}$ [Het], $1.2\pm0.2 \times 10^{-3}$ [KO]; Fig 2A and 2B), indicating defective spermatogenesis in *Trim41* KO testis. To define the cause of testicular atrophy, we performed hematoxylin and periodic acid-Schiff (HePAS) staining of testicular sections and found that the germ cell layer was thinner in *Trim41* KO testis than in control testis (Fig 2C; low magnification). Then we compared testicular cells based on the cycle of the seminiferous epithelium [21,22]. From this analysis, we determined the population of the first germ cell layer containing pleleptotene stage (Fig 2C; stage VII–VIII) and zygotene stage (Fig 2C; stage XII) was comparable between Het and KO testis. On the other hand, pachytene stage (Fig 2C; stage VII–VIII) and diplotene stage (Fig 2C; stage XII) spermatocytes and spermatids were dramatically decreased in the KO testis. Although a few elongating/elongated spermatids were present in the KO testis, their nuclei were not fully compacted (Fig 2C; red arrows). Consistent with the dramatic decrease of spermatocytes and spermatids, the number of TUNEL positive cells increased in KO testis compared with Het counterparts (Fig 2D and 2E). The TUNEL positive cells were frequently observed in the second germ cell layer containing spermatocytes after the pachytene stage (Fig 2D; high magnification). We did not see the accumulation of TUNEL positive cells in a specific stage of the seminiferous epithelium cycle (Fig 2E). These results suggested that spermatocytes after the pachytene stage might failed in meiotic processes [23,24]. Consistent with the results in adult testis, the weight decline of juvenile testis and apoptotic elimination of germ cells became prominent from PND16, corresponding to when the first wave of spermatogenesis reaches the pachytene stage (S3A and S3B Fig). As a result of spermatogenesis defects in *Trim41* KO testis, only a few mature spermatozoa were present in the cauda epididymis (Fig 2F). In addition, all the spermatozoa exhibited abnormal head/tail shapes (S4A–S4D Fig) and were immotile (S4E Fig). These *Trim41* KO spermatogenesis defects were restored by *Clgn*-driven *Trim41* (Fig 2A–2D), albeit partially by TUNEL analysis (Fig 2E). These observations suggested that *Trim41* KO males exhibited spermatogenesis defects, leading to oligozoospermia.

## *Trim41* KO spermatocytes underwent SYCP3 overloading

Due to an apparent defect during meiosis, we examined DNA DSBs and synapsis by immunostaining surface chromosome spreads with anti-γH2AX and SYCP3 antibodies, respectively (Fig 3A and 3B). The SYCP3/γH2AX immunostaining pattern in leptotene and zygotene stage spermatocytes was comparable between the two genotypes, showing that *Trim41* KO spermatocytes underwent programmed DSBs and initial assembly of the synaptonemal complex. However, the compaction of the XY axes, a telltale signature of XY body formation, was rarely observed in KO spermatocytes (Fig 3B–3D; spermatocytes exhibiting XY body malformation: 6/181 [Het], 192/207 [KO]). Consistent with the XY body malformation, the pachytene and diplotene populations decreased dramatically in KO testis (Fig 3E). More strikingly, intense SYCP3 signals were observed on both autosomes and sex chromosomes (Fig 3B–3D; spermatocytes with intense SYCP3 signals: 3/181 [Het], 205/207 [KO]). Such overloading of SYCP3 became evident once homolog synapsis had been completed at pachytene rather than at the zygotene/pachytene transition (zygotene/pachytene transition; S5A and S5B Fig). Furthermore, meticulous observation revealed a separation of SYCP3 signals in autosomes ([Fig

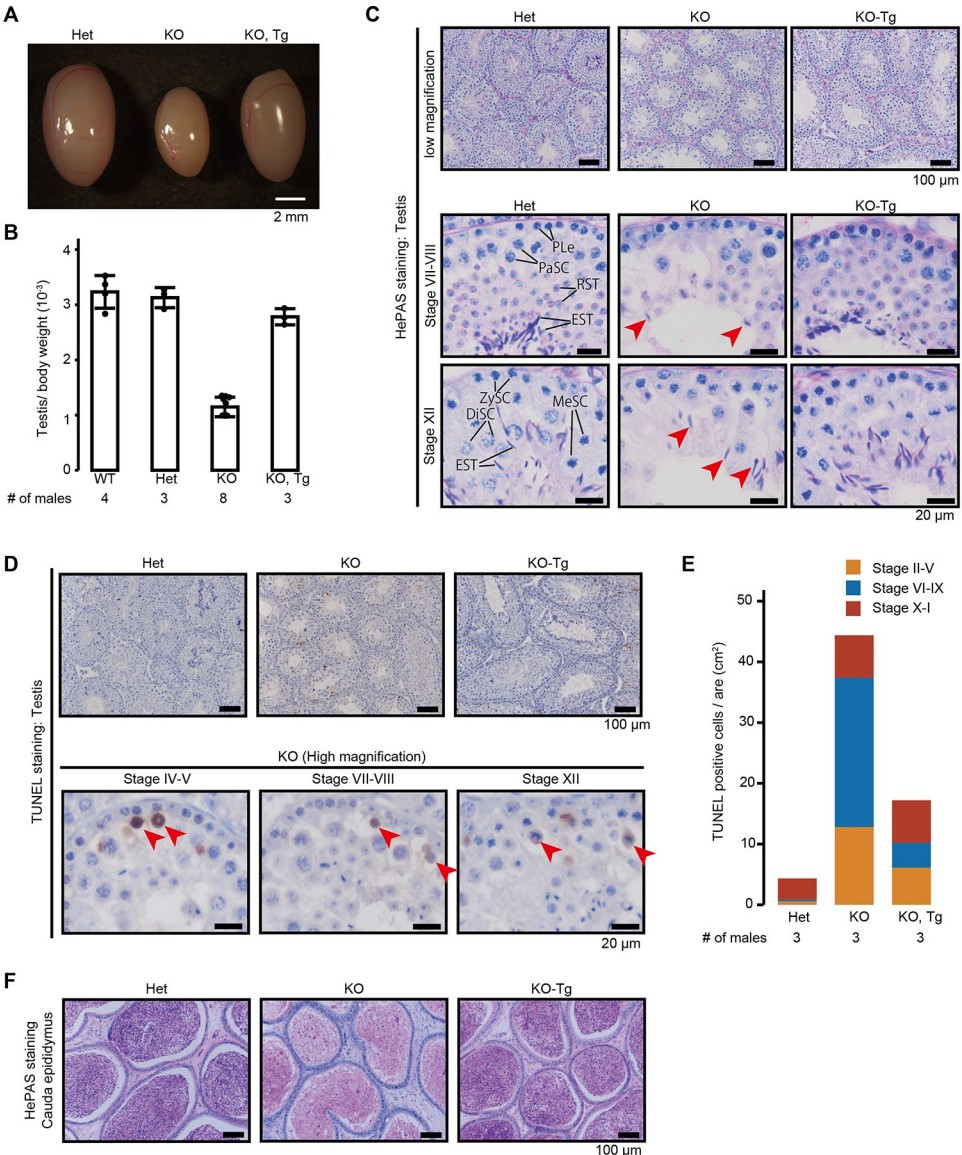

**Fig 2. Histological analysis of *Trim41* KO male mice.** (A and B) Testis morphology (A) and testis/bodyweight of WT, *Trim41* Het, *Trim41* KO, and KO-Tg adult mice (B). Error bars indicate standard deviation. The numerical data are available in S3 Table (C) PAS staining of seminiferous tubules of adult mice. The seminiferous epithelium cycle was determined by germ cell position and nuclear morphology. Red arrows indicate elongated spermatids with abnormal head morphology. PLe: Preleptotene stage spermatocyte; ZySC: Zygotene stage spermatocyte; PaSC: Pachytene stage spermatocyte; DiSC: Diplotene stage spermatocyte; MeSC: metaphase spermatocyte; RST: round spermatid; EST: elongating/elongated spermatid. (D) TUNEL staining of seminiferous tubules of adult mice counterstained with hematoxylin. At least three male mice were analyzed. (E) The stacked bar graph of TUNEL positive cells. The seminiferous epithelial stages were roughly determined by the arrangement and nuclear morphology of the first layer of germ cells (spermatogonia and leptotene/zygotene stage spermatocytes). The numerical data are available in S3 Table (F) PAS staining of cauda epididymis of adult mice. At least three male mice were analyzed.

3C; white arrowheads), two SYCP3 axes on sex chromosomes (Fig 3C; yellow arrowhead), and tangled SYCP3 signals between the sex chromosome and autosomes (Fig 3C; red arrowhead). We did not observe apparent abnormalities other than in SYCP3-overloaded regions. Of note, the centerline with intense SYCP3 signals showed weak γH2AX and DNA staining (S5C Fig;

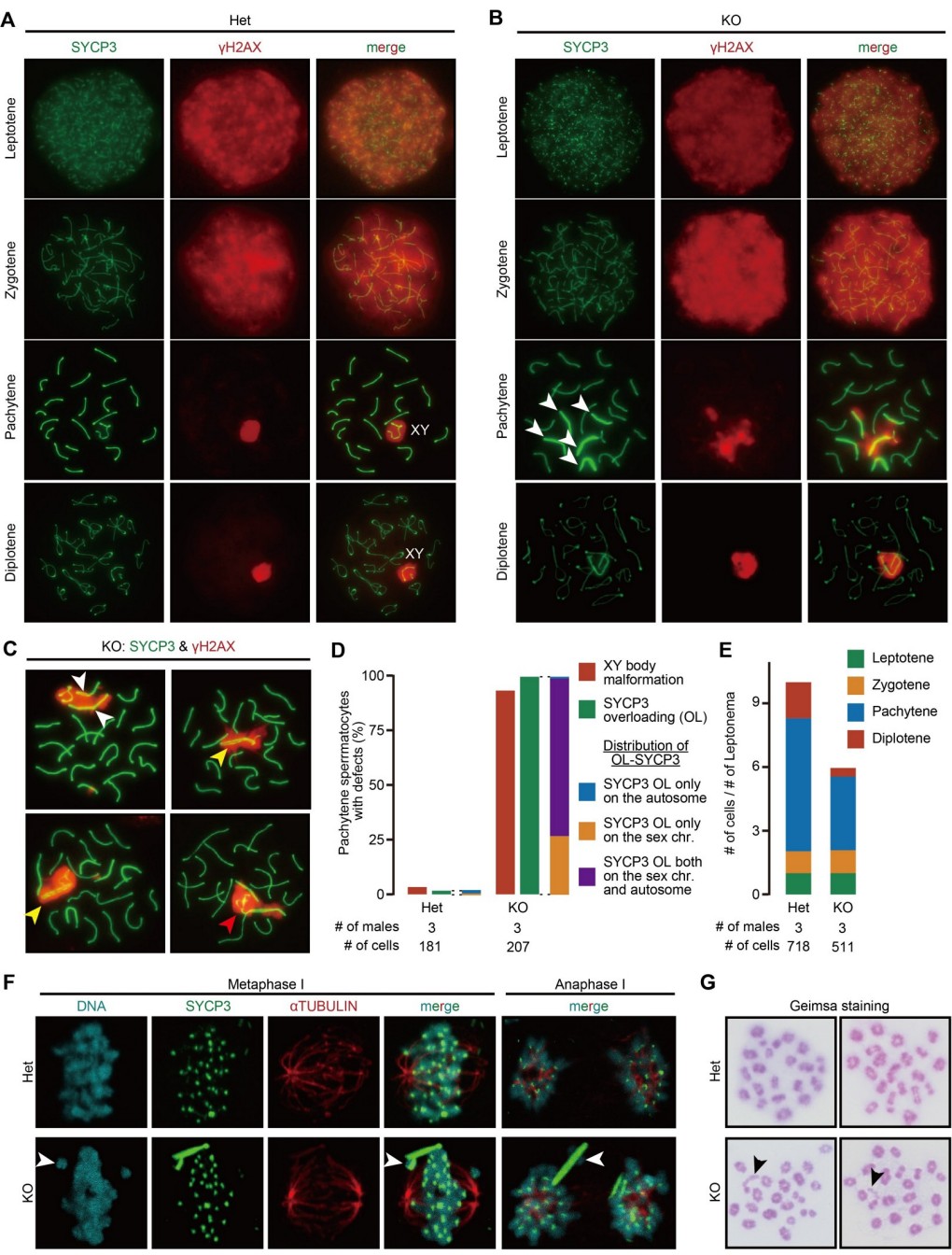

**Fig 3. Cytological analysis of *Trim41* KO spermatocytes.** (A and B) The spread nuclei of prophase spermatocytes collected from adult Het (A) and KO (B) male mice were immunostained with anti-SYCP3 and -γH2AX antibodies. XYs in (A) indicate the sex chromosomes encircled by γH2AX signals (i.e. XY body). White arrowheads in (B) indicate intense SYCP3 signals. At least three male mice were analyzed. (C) Additional immunostaining images of KO spermatocytes. The white arrowheads indicate the separate SYCP3 signals in autosomes. The yellow arrowhead indicates the X chromosome with two SYCP3 axes. The red arrowhead shows a tangled SYCP3 signal between the sex chromosome and autosomes (only 17 axes existed other than the tangled SYCP3 signals). (D) The percentage of pachytene stage spermatocytes exhibiting XY body malformation (red bar graph) and SYCP3 overloading (green bar graph). Blue, purple, and yellow-colored boxes show the distribution of chromosomes (i.e., sex chromosome or autosome) exhibiting SYCP3 overloading. The numerical data are available in S3 Table (E) The percentage of each meiotic prophase stage in immunostained spread nuclei samples in A and B. The numerical data is available in S3 Table (F) Immunostaining of metaphase and anaphase spermatocytes squashed from seminiferous tubules after fixation. Blue arrowheads indicate chromosomes connected with the rod-like SYCP3 structures. At least three male mice were

analyzed. The numerical data are available in S3 Table (G) Giemsa staining of spread nuclei of metaphase I spermatocytes. Black arrows indicate the chromosomes with a rod-like structure. At least three male mice were analyzed. The numerical data are available in S3 Table.

white arrowheads), suggesting some fraction of the overloaded SYCP3 might not associate with chromosomes.

Since localization of axial element proteins, SYCP2 and SYCP3, depends on the cohesin axial core that REC8 and RAD21L generate [25–27], we examined REC8 (S6A and S6B Fig) and RAD21L (S6C and S6D Fig). However, the localization pattern and staining signal of REC8 and RAD21L were comparable between Het and KO spermatocytes.

The overloaded SYCP3 remained even in metaphase I and anaphase I (Fig 3F; spermatocytes with rod-like SYCP3 signals: 0/308 [Het, metaphase I], 327/341 [KO, metaphase I], 0/50 [Het, anaphase I], 29/30 [KO, anaphase I]). In addition, some chromosomes connected with SYCP3 failed to align at the equatorial plate and were left behind between two centrosomes (Fig 3F; white arrowheads; observed chromosome mislocation: 2/308 [Het, metaphase I],33/ 341 [KO, metaphase I], 1/50 [Het, anaphase I], 5/30 [KO, anaphase I]). Probably because of the remaining SYCP3 axes, one or two chromosomes exhibited a rod-like structure at metaphase (Fig 3G; black arrows; spermatocytes with rod-like chromosomes: 0/112 [Het]; 106/130 [KO]). These results showed that the SYCP3 dynamics on chromosome axes was misregulated in *Trim41* KO spermatocytes.

## SYCP3 overloading was biased to the X chromosome in male meiosis

To examine whether SYCP3 overloading is biased to specific chromosomes, we combined SYCP3 immunostaining with multicolor fluorescence in situ hybridization (mFISH; Fig 4A and 4B). The result showed the X chromosomes with more frequent SYCP3 overloading than autosomes (Fig 4C; X chromosome: 37 out 43 cells; autosome: 12 out 43 cells), although we might have missed minor abnormalities of SYCP3 signals because of harsh conditions of FISH procedure. Of note, the autosomes with overloaded SYCP3 signals appeared random (Fig 4C). As the most affected chromosome, the X chromosome, is also present in female cells, we examined female pachytene germ cells in E15.5 ovaries. The gross ovary morphology was comparable between the two genotypes (S7A Fig). SYCP3 overloaded axes were observed only in a few female germ cells (S7B Fig). This result contrasts the observation that SYCP3 overloading was frequent on the X chromosome in spermatocytes (Fig 4C). As a result, folliculogenesis progressed normally (S7C Fig), and *Trim41* KO female mice were fully fertile (Fig 1M). Overall, these results showed that *Trim41* is essential for regulating SYCP3 protein loading, especially on the X chromosome during male meiosis.

## SYCP3 overloaded X chromosomes exhibited an abnormal synapsis state

To examine the synapsis state, we visualized the central element of the synaptonemal complex (synapsed axes; Fig 5A and 5B) and unsynapsed axes (Fig 5C and 5D) by immunostaining surface chromosome spreads with anti- SYCP1 and BRCA1 antibodies, respectively. SYCP1 is a molecule known to localize between the lateral elements of the synaptonemal complex of homologous chromosomes and form transverse filaments, akin to rungs of a ladder during synapsis [28,29]. BRCA1, the product of breast cancer 1, accumulates on unsyapsed axes of sex chromosomes [30].

The immunostaining showed that SYCP1 on the sex chromosomes tended to remain in KO (Figs 5A, 5B and 5E; pachytene spermatocytes with SYCP1 signals outside of the pseudoautosomal region of sex chromosomes (Blue arrowheads): 190/359 [Het, early pachytene], 9/120

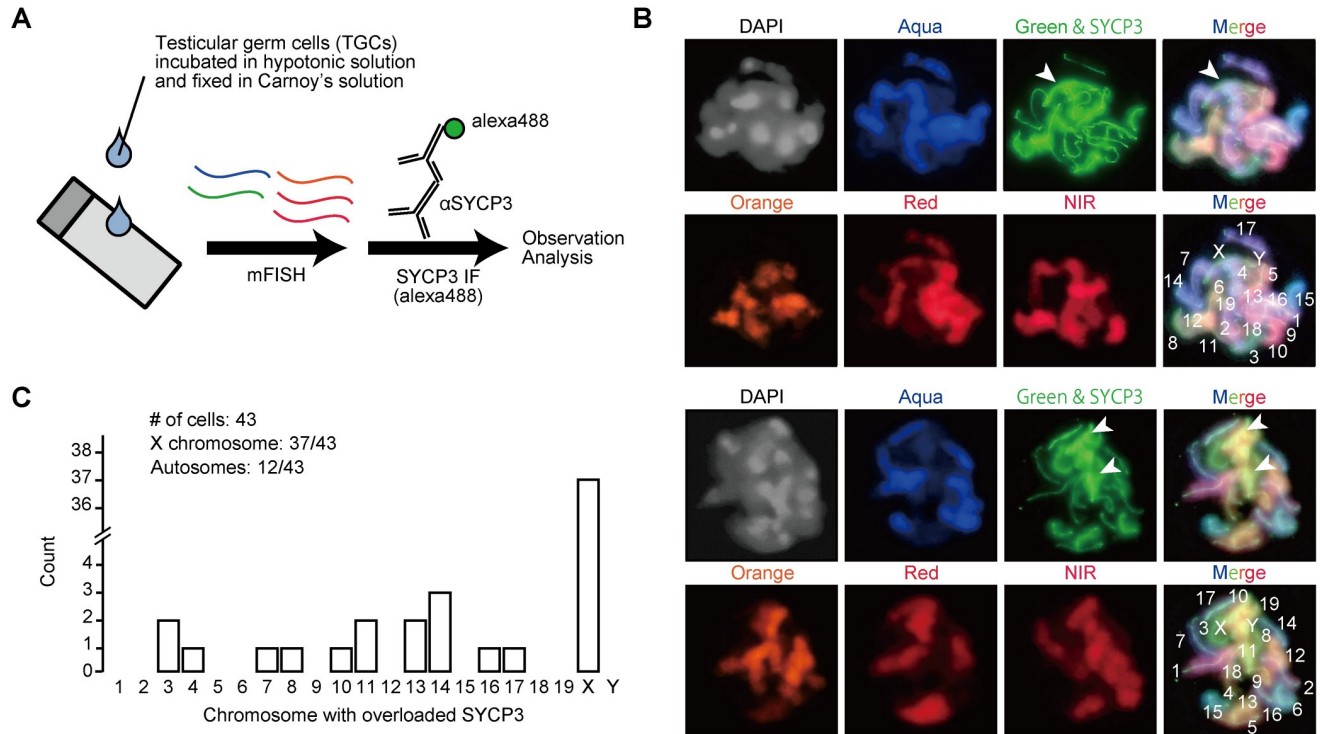

**Fig 4. Cytological analysis of *Trim41* KO spermatocytes.** (A) Schematic of multicolor FISH followed by SYCP3 immunostaining. (B) Pachytene stage spermatocytes were subjected to a multicolor FISH protocol shown in A. Green FISH probe and Alexa-488 are detected by the same filter. White arrowheads indicate overloaded SYCP3 signals. (C) The histogram for the chromosome distribution of SYCP3 overloading. Forty-three well-spread cells were analyzed.

[Het, late pachytene], 206/254 [KO, early pachytene], 64/102 [KO, late pachytene]). Furthermore, we observed seamlessly-connected and tangled SYCP1/SYCP3 signals between the sex chromosomes and autosomes (Figs 5B, S8A–S8C; seamless connection of SYCP1/SYCP3 signals (white arrowheads): 1/70 [Het], 9/41 [KO]; tangled SYCP1/SYCP3 singles (white asterisks): 0/70 [Het], 9/41 [KO]). On the other hand, the BRCA1 staining pattern became uneven throughout the X chromosomes (Fig 5D and 5F; uneven BRCA1 staining on sex chromosomes (yellow arrowheads): 5/138 [Het], 127/154 [KO]). Furthermore, BRCA1 was positive on several autosomes with overloaded SYCP3 (Fig 5D and 5F; BRCA1 locating on autosomes (green arrowheads): 2/138 [Het], 41/154 [KO]).

Since HORMA domain-containing proteins localize to unsynapsed axes and function in the quality surveillance of homolog alignment and activation of the DNA damage response [31,32], we examined HORMAD1 localization by immunostaining (S9A and S9B Fig). Consistent with BRCA1 immunostaining results, HORMAD1 remained on autosomes with overloaded SYCP3 (S9B Fig; white arrowheads). These defects in SYCP1, BRCA1, and HORMAD1 localization patterns were observed only in SYCP3-overladed regions. These results suggested that SYCP3 overloaded areas in *Trim41* KO spermatocytes underwent an abnormal synapsis configuration.

## *Trim41^ΔRING/ΔRING^* male mice phenocopied *Trim41^neo/neo^* male mice

To examine whether the RING domain is essential for *Trim41* function, we disrupted the RING domain by inserting the HA affinity tag sequence (Fig 6A; referred to as ΔRING). The ΔRING-TRIM41 loses only ubiquitin ligase activity, not interaction with target proteins [9,10].

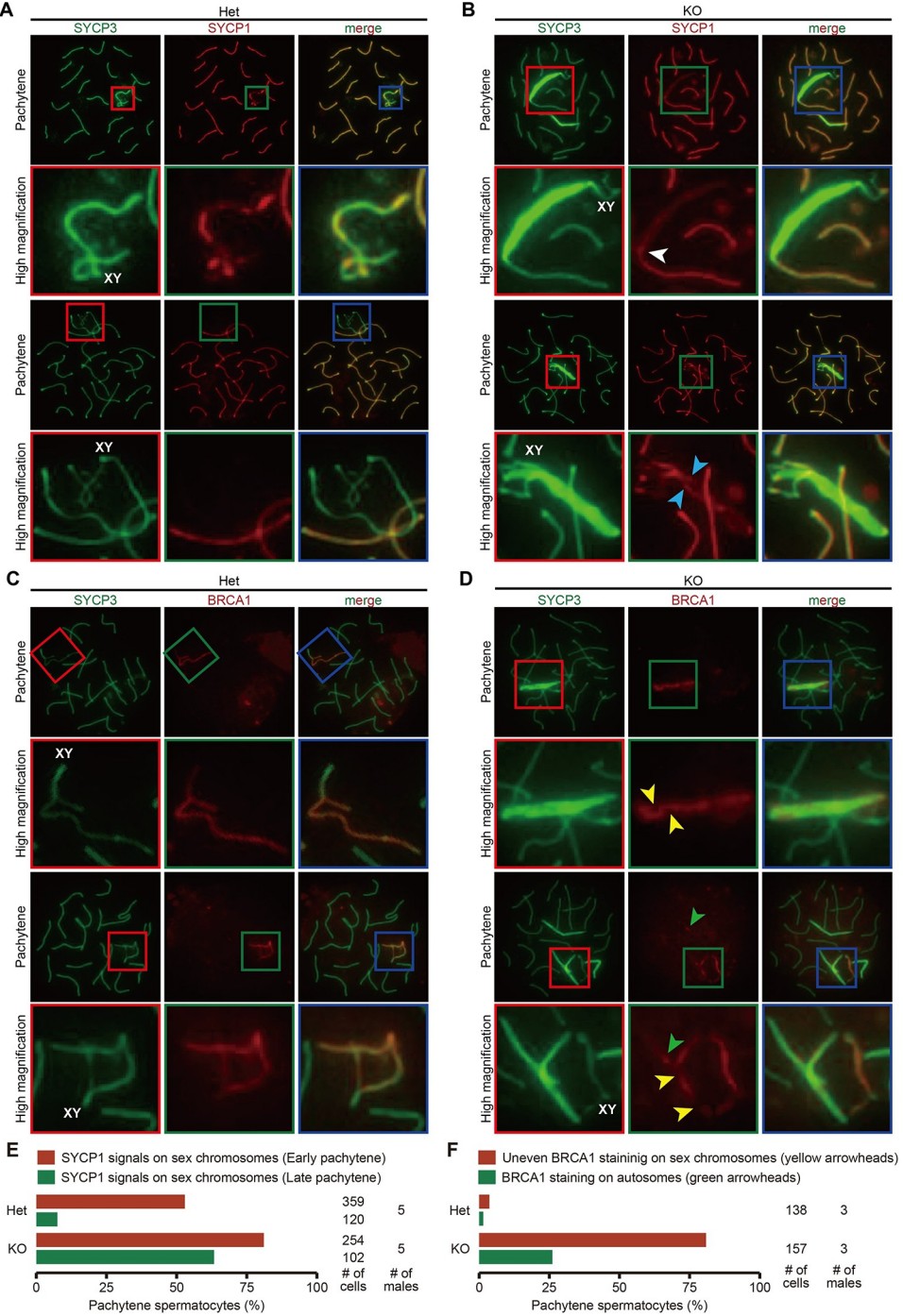

**Fig 5. Synapsis configuration of SYCP3 overloaded axes.** (A and B) SYCP1 immunostaining of spread nuclei of prophase spermatocytes collected from adult Het (A) and KO (B) male mice. High magnified images for the sex chromosomes are shown in red, green, and blue boxes. The sex chromosomes were identified by their characteristic shape of SYCP3 axes. A white arrowhead indicates a seamless connection of SYCP1 signals between the sex chromosomes and autosomes. Blue arrowheads show SYCP1 signals remaining on the sex chromosomes until the late pachytene stage. At least three male mice were analyzed. (C and D) BRCA1 immunostaining of spread nuclei of prophase spermatocytes collected from adult Het (C) and KO (D) male mice. High magnified images for the sex chromosome were shown in red, green, and blue boxes. Yellow arrowheads indicate BRCA1 missing from the sex chromosome. Green arrowheads show BRCA1 located on autosomes. At least three male mice were analyzed. (E) Quantification data for SYCP1 staining. Red and green bar graphs indicate the percentage of early and late pachytene spermatocytes with SYCP1 signals outside of the pseudoautosomal region of sex chromosomes, respectively. The

numerical data is available in S3 Table (F) Quantification data for BRCA1 staining. Red bar graph indicates the percentage of pachytene spermatocytes exhibiting uneven BRCA1 signals on the sex chromosome. Green bar graph shows the percentage of pachytene spermatocytes with BRCA1 signals on the autosomes. The numerical data are available in S3 Table.

We microinjected two gRNA/Cas9 ribonucleoprotein complexes and a ssODN into zygotes (Fig 6A). Of 145 injected eggs, 119 eggs reached the two-cell stage. Then we transplanted the two-cell eggs into the oviducts of six pseudopregnant female mice and obtained twenty-five pups. Genotype PCR screening (Fig 6B) detected the intended mutation with a 180 bp deletion and HA tag insertion (referred to as $Trim41^{\Delta RING}$) in ten pups. RT-PCR analysis confirmed the gene expression in $Trim41^{\Delta RING/\Delta RING}$ testis, albeit not in $Trim41^{neo/neo}$ counterparts (Fig 6C). Immunoblotting analysis with anti-HA and -TRIM41 antibodies detected HA-tagged ΔRING-TRIM41 expression (Fig 6D and 6E). Then, we caged $Trim41^{\Delta RING/\Delta RING}$ male mice with WT females for two months to analyze their fertility. Although we confirmed 38 vaginal plugs, $Trim41^{\Delta RING/\Delta RING}$ males failed to sire any pups (Fig 6F). Further, the testis of $Trim41^{\Delta RING/\Delta RING}$ males was smaller than the $Trim41^{wt/\Delta RING}$ counterparts (Fig 6G and 6H) due to spermatogenesis defects (Fig 6I). TUNEL-positive cells were frequently observed in the 2nd germ cell layer of seminiferous tubules (Fig 6J), corresponding to spermatocytes after the pachytene stage in $Trim41^{\Delta RING/\Delta RING}$ mice. Also, TUNEL signals overall were positive throughout the seminiferous epithelial cycles (Fig 6K). As a result, almost no fully-matured spermatozoa existed in the cauda epididymis of $Trim41^{\Delta RING/\Delta RING}$ males (Fig 6L). These results showed that $Trim41^{\Delta RING/\Delta RING}$ male mice exhibited the same phenotype as $Trim41^{neo/neo}$ male mice, suggesting that the RING domain is essential for TRIM41 function.

## ΔRING-TRIM41 accumulated on SYCP3-overloaded region

Immunostaining analysis with anti-SYCP3 and γH2AX antibodies demonstrated XY body malformation and SYCP3 overloading in $Trim41^{\Delta RING/\Delta RING}$ pachytene stage spermatocytes (Fig 7A and 7B). Also, the number of pachytene and diplotene spermatocytes decreased dramatically in $Trim41^{\Delta RING/\Delta RING}$ testis (S10A–S10C Fig). These results corroborated the necessity of the RING domain in TRIM41 functions.

Next, to determine ΔRING-TRIM41 localization, we stained the spread nuclei with an anti-HA antibody (Fig 7C–F). In $Trim41^{wt/\Delta RING}$ spermatocytes, we observed only faint signals throughout the nuclei (Fig 7C). However, surprisingly, $Trim41^{\Delta RING/\Delta RING}$ spermatocytes saw intense HA signals on the SYCP3-overloaded regions (Figs 7D and S11A; white arrowheads). The strong HA signals on SYCP3 appeared after the zygotene–pachytene stage transition (S11A Fig). In addition, the HA signals tended to become stronger during prophase I progression (S11A Fig), showing a correlation with SYCP3 overloading. The HA foci were also detected in $Trim41^{neo/\Delta RING}$ spermatocytes (Fig 7E), demonstrating that the removal of the RING domain changed the immunostaining pattern, not the protein level difference. On the other hand, we did not detect the HA immunostaining signals in *Trim41* KO spermatocyte with SYCP3 overloading (Fig 7F), ruling out the possibility of antibody cross-reaction. Furthermore, we confirmed the same phenomena in immunostaining analysis with an anti-TRIM41 antibody (S11B–S11E Fig). These observations showed that ΔRING-TRIM41 accumulated on the SYCP3-overloaded regions, suggesting that TRIM41 functions on the chromosome axes. As a side note, we also identified several HA foci outside of the chromosome axes in zygotene and early pachytene stage spermatocytes (Figs 7D and S11A; blue arrowheads), although we could not specify the location of those foci.

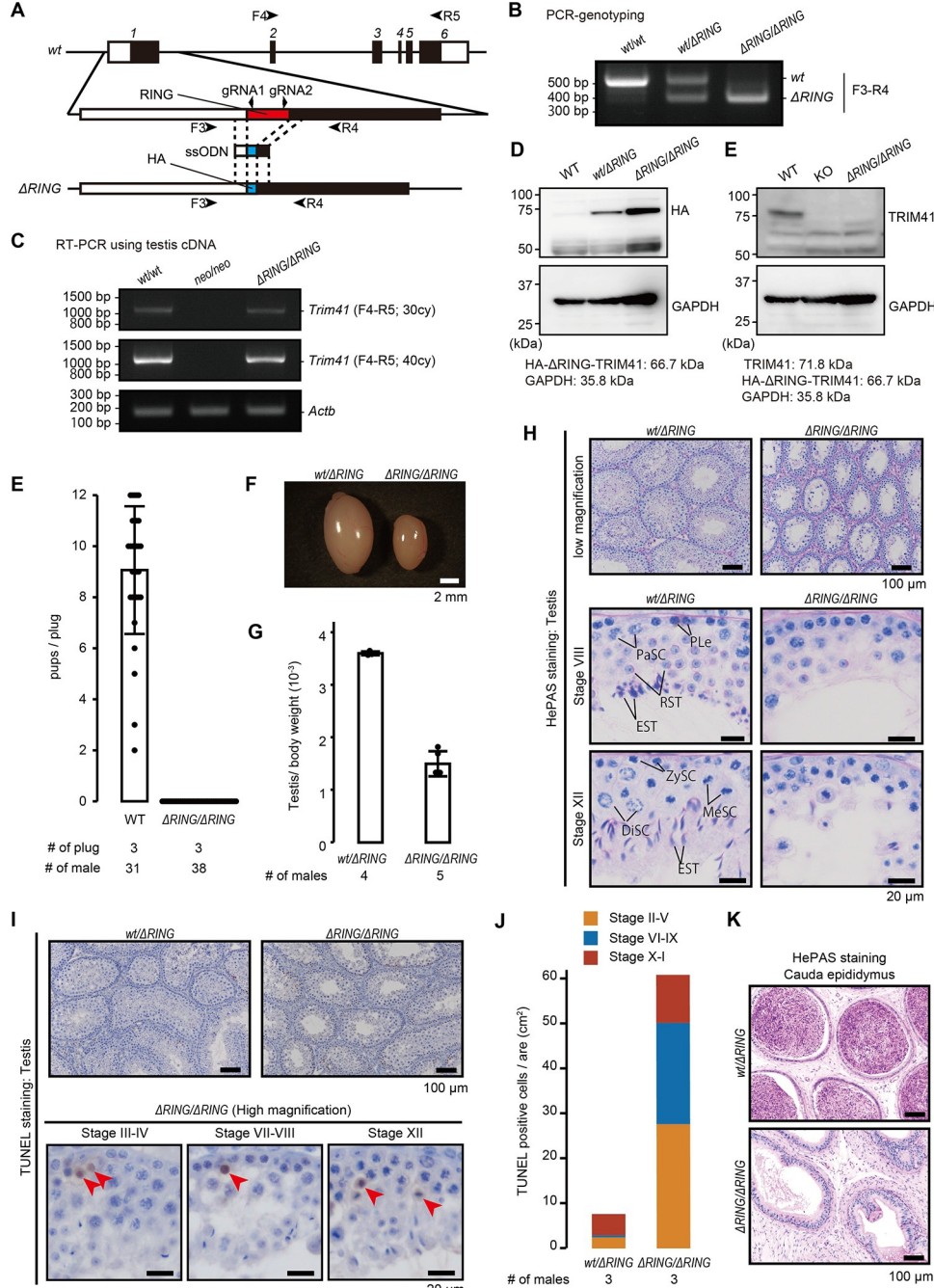

**Fig 6. Production and phenotypic analysis of *Trim41^{ΔRING/ΔRING}* mice.** (A) Knock-in scheme for the replacement of the RING domain with the HA tag. Black and white boxes in the gene map indicate coding and non-coding regions, respectively. Black arrowheads (F3, R4, F4, and R5) and arrows (gRNA1 and gRNA2) indicate primers for genotyping and target sequence of gRNAs, respectively. (B) An example of genotyping PCR with primer sets in A (F3 and R4). (C) RT-PCR using testis cDNA and primer sets in A (F4 and R5). *Actb* was used as a loading control. (D and E) Immunoblotting analysis with anti-HA (D) and -TRIM41 (E) antibodies. Black and red arrows indicate TRIM41 andΔRING-TRIM41, respectively. (F) The result of mating tests. Pups/plug: 9.1±2.5 [WT]; 0 [*Trim41^{ΔRING/ΔRING}*] (s. d.). Error bars indicate standard deviation. The numerical data are available in S3 Table (G and H) Testis morphology (G) and testis/bodyweight of *Trim41^{wt/ΔRING}* and *Trim41^{ΔRING/ΔRING}* adult mice (H). Error bars indicate standard deviation. The numerical data are available in S3 Table (I) PAS staining of seminiferous tubules of adult mice. The seminiferous epithelium cycle was determined by germ cell position and nuclear morphology. PLe: Preleptotene stage spermatocyte; ZySC: Zygotene stage spermatocyte; PaSC: Pachytene stage spermatocyte; DiSC: Diplotene stage spermatocyte; MeSC: metaphase spermatocyte; RST: round spermatid; EST: elongating/elongated spermatid. At least

three male mice were analyzed. (J) TUNEL staining of seminiferous tubules of adult mice counterstained with hematoxylin. At least three male mice were analyzed. (K) The stacked bar graph of TUNEL positive cells. The seminiferous epithelial stages were roughly determined by the arrangement and nuclear morphology of the first layer of germ cells (spermatogonia and leptotene/zygotene stage spermatocytes). The numerical data are available in S3 Table (L) PAS staining of cauda epididymis of adult mice counterstained with hematoxylin. At least three male mice were analyzed.

## ΔRING-TRIM41 did not colocalize with recombination nodules

The spotty immunostaining pattern of TRIM41 is reminiscent of the distribution of recombination nodules such as RAD51, DMC1, MSH4, and MLH1. Therefore, we examined the DMC1 (S12 Fig) and MLH1 (S13 Fig) localization pattern by immunostaining. DMC1, a meiosis-specific recombination protein, localizes to the resected ssDNA-ends and carries out a homology search [33–35]. Although the number of DMC1 foci per nucleus peaks at the zygotene stage and decreases during pachytene progression (S12A–S12C Fig), the DMC1 foci remained in late pachytene and diplotene stage KO spermatocytes (S12B–S12C Fig). The remaining DMC1 foci tended to associate with SYCP3-overloaded regions (S12D Fig). However, DMC1 foci did not colocalize with ΔRING-TRIM41, suggesting that the remaining DMC1 might be a downstream defect.

We next examined the MLH1 foci, a DNA-mismatch protein promoting cross-over during meiosis [36,37]. The number of MLH1 foci per nucleus was comparable between Het and KO spermatocytes (S13A–S13C Fig). We did not observe either the specific association pattern between MLH1 foci and overloaded SYCP3 (S13D Fig) or colocalization of MLH1 foci and ΔRING-TRIM41 (S13E Fig). These results suggested that TRIM41 did not directly regulate recombination protein dynamics.

## ΔRING-TRIM41 accumulated on chromosome axes exhibiting an abnormal synapsis state

We examined the relationship between ΔRING-TRIM41 accumulation and the synapsis state by SYCP1 and BRCA1 immunostaining. Consistent with the result in KO mice, SYCP1 tended to remain on the sex chromosome axes in *Trim41*$^{ΔRING/ΔRING}$ spermatocytes, although the sex chromosome SYCP1 axes were uneven (Figs 7G and S14A). Then, we compared the SYCP1 immunostaining pattern with the ΔRING-TRIM41 accumulation pattern and noticed that the boundary of the SYCP1 positive and negative parts were hotspots of ΔRING-TRIM41 accumulation (Figs 7G and S14A; magenta arrowheads). On the other hand, in BRCA1/HA immunostaining analysis, we observed ΔRING-TRIM41 on the BRCA1 negative part of the X chromosome axes (Figs 7H and S14B; red arrowheads). Taken together, these observations suggested that ΔRING-TRIM41 accumulated on the synapsed part of the X chromosome axes, and therefore TRIM41 is essential for preventing the abnormal synapsis configuration of the X chromosome.

## Substrate candidates were not identified

Given that TRIM41 confers ubiquitin ligase E3 activity [9,10], we examined ubiquitin localization along chromosome axes [38]. However, the ubiquitin localization pattern was comparable between Het and KO spermatocytes (S15A and S15B Fig), showing the overall ubiquitination activity on chromosome axes was not affected by TRIM41 disruption. Furthermore, although we tried to identify the substrates by immunoprecipitation, we could not either detect SYCP3 in co-IPed eluate samples by immunoblotting (S16A–S16F Fig) or find candidate proteins in IP-Mass analysis (S16G and S16H Fig). The protein amount and ubiquitination level of several

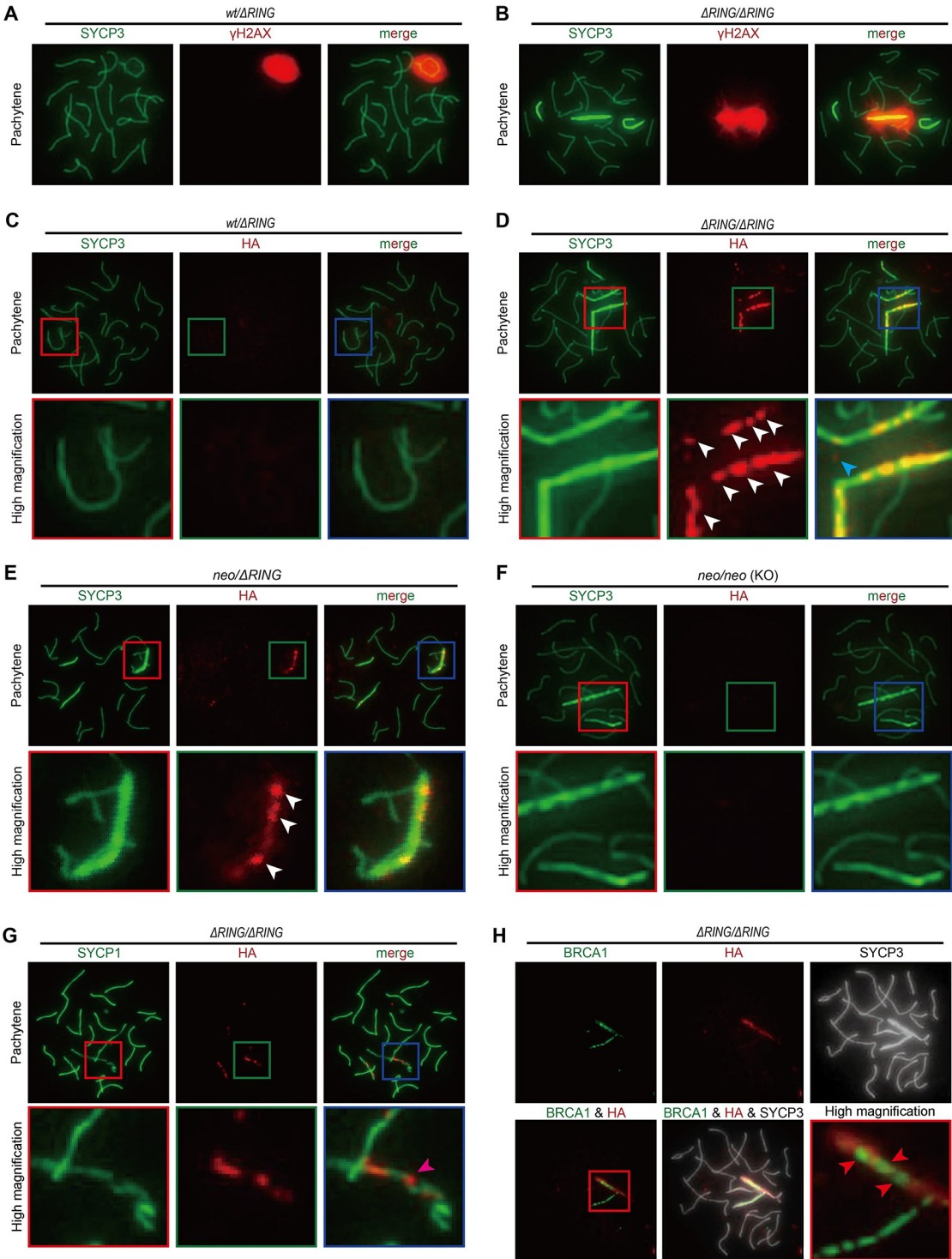

**Fig 7. Cytological analysis of *Trim41^{ΔRING/ΔRING}* spermatocytes.** (A and B) SYCP3/γH2AX immunostaining of pachytene stage spermatocytes from *Trim41^{wt/ΔRING}* (A) and *Trim41^{ΔRING/ΔRING}* (B) male mice. At least three male mice were analyzed. (C–F) SYCP3/HA immunostaining of pachytene spermatocytes from *Trim41^{wt/ΔRING}* (C), *Trim41^{ΔRING/ΔRING}* (D), *Trim41^{neo/ΔRING}* (E), and *Trim41^{neo/neo}* (KO; F) male mice. White and blue arrowheads indicate HA signals on and outside of SYCP3 axes, respectively. High magnified images are shown in red, green, and blue boxes. At least three male mice were analyzed for *Trim41^{wt/ΔRING}* and *Trim41^{ΔRING/ΔRING}*. (G) SYCP1/HA immunostaining of surface chromosome spread from *Trim41^{ΔRING/ΔRING}* testis. Magenta arrowheads indicate HA signals on the SYCP1 positive/negative boundary. High magnified images are shown in red, green, and blue boxes. At least three male mice were analyzed. (H) BRCA1/HA immunostaining of surface chromosome spread from

*Trim41*$^{\Delta RING/\Delta RING}$ testis. Red arrowheads indicate HA signals on the BRCA1 negative part of the X chromosome axes. A high magnified image was shown in a red box. At least three male mice were analyzed.

axis-related proteins, including SYCP3, SYCP1, HORMAD1, were comparable between Het and KO testis (S16I and S16J Fig). Thus, TRIM41 substrate discovery would be an area of feature research.

## Discussion

In this research, we produced *Trim41* KO mice and analyzed the phenotype. Unexpectedly, *Trim41* deficiency caused meiosis defects and infertility in male mice despite ubiquitous expression. The infertility was rescued by the transgene driven by a testis-specific *Clgn* promoter. In detailed analyses, we found that *Trim41* KO spermatocytes exhibited SYCP3 overloading. Furthermore, we removed the RING domain of TRIM41 (ΔRING) to examine the necessity of the RING domain and found that *Trim41*$^{\Delta RING/\Delta RING}$ phenocopied *Trim41*$^{neo/neo}$ (i.e., KO). Surprisingly, the ΔRING-TRIM41 accumulated on the SYCP3-overloaded regions.

The most striking phenotype of *Trim41* KO spermatocytes was SYCP3 overloading. In addition, SYCP1 tended to remain on the SYCP3-overloaded X chromosome, suggesting self-synapsis of the X chromosome. Although we could not specify, this self-synapsis might be inter-sister synaptonemal complex assembly like those found in REC8 KO spermatocytes [26,39,40]. However, given that we also observed multi- SYCP1 axes in the X chromosome (Fig 5B; see blue arrowheads), we should also consider the possibility of extensive heterologous self-synapsis like those seen in domestic dog pachytene spermatocytes [41]. The latter possibility is corroborated by another outstanding phenotype of the seamlessly-connected and tangled SYCP1/SYCP3 signals between the X chromosome and autosomes. The XY body [13] malformation phenotype also indicated that *Trim41* KO spermatocytes failed in the physical separation of their sex chromosome. Further studies are needed to determine the relationship between these two phenotypes (i.e., SYCP3 overloading and abnormal synapsis configuration; causal relationship or simultaneous phenomena) and whether these phenotypes are direct influences of TRIM41 deficiency or compensatory changes.

To examine which chromosomes exhibited SYCP3 overloading in *Trim41* KO spermatocytes, we performed multicolor FISH combined with SYCP3 immunostaining and found that the X chromosome was mainly affected. On the other hand, autosomes saw fewer SYCP3 overloading with no bias to specific chromosomes. A possible reason for the X chromosome bias is the unsynapsed nature during the pachytene stage. This inference is in agreement with the fact that only a tiny population of germ cells underwent SYCP3 overloading in *Trim41* KO females, whose X chromosomes have fully synapsed during prophase I progression. Of course, we acknowledge that the weak phenotype in females might be due to the physiological difference of male and female meiosis. Therefore, *Trim41*-deficient XO meiosis (i.e. mouse model of Turner Syndrome) might be an intriguing future consideration.

To examine the molecular function of TRIM41, we also produced *Trim41*$^{\Delta RING/\Delta RING}$ mice by replacing 2–61 aa residues with HA-tag sequence, and we found that the HA immunostaining signals on the overloaded-SYCP3 region. As the HA signals appeared only when the cells had no functional TRIM41 (i.e. *Trim41*$^{\Delta RING/\Delta RING}$ or *Trim41*$^{neo/\Delta RING}$), we concluded that ΔRING-TRIM41 accumulated on the SYCP3-overloaded regions. This observation suggested that ΔRING-TRIM41 kept targeting the unubiquitinated substrates, and therefore TRIM41 exerts its ubiquitin ligase E3 activity on the chromosome axes. This ubiquitin ligase E3 activity of TRIM41 might be a possible mechanism of axis dynamics misregulation and SYCP3 overloading in KO spermatocytes. However, we could not narrow down the substrate candidate by

IP-Mass analysis. Also, we did not see an interaction between TRIM41 and SYCP3. As the weak/transient binding between enzymes and substrates tends to be challenging to capture in IP analysis [42], the recently developed proximity-dependent biotin identification (BioID) technique [43,44] could be a future consideration in substrate identification. Another possible mechanism of SYCP3 overloading in *Trim41* KO is that SYCP3 was misassembled into an alternative structure, which does not associate with chromosomes [45,46]. Therefore, ultra-structure observation by electron microscopy would be another future consideration.

In summary, our result showed that *Trim41* is essential for preventing SYCP3 overloading. Further studies for substrate identification will unveil the exact molecular functions of TRIM41 and the mechanism for chromosome axes remodeling.

## Material & methods

### Ethics statement

All animal experiments were approved by the Animal Care and Use Committee of the Research Institute for Microbial Diseases, Osaka University (#Biken-AP-R03-01-0).

### Animals

Animals were housed in a temperature-controlled environment with 12 h light cycles and free access to food and water. B6D2F1 (C57BL/6 × DBA2; Japan SLC, Shizuoka, Japan) mice and ICR (SLC) were used as embryo donors; B6D2F1 were used for mating and wild-type control; C57BL6/N (SLC) mice were used to collect RNA for cloning.

### Generation of Trim41 KO mice

To generate *Trim41* KO mice, we replaced exon1 with a neomycin resistance (Neo) cassette in EGR-G101 ES cells [47] by transfecting a targeting vector. The short and long arm corresponds to chr11: 48,707,466–48,708,476 and chr11: 48,697,891–48,706,384 (GRCm39), respectively. Potentially targeted ES cell clones were selected with neomycin. Correctly targeted ES cell clones and germ-line transmission were confirmed via PCR using primers (GeneDesign, Osaka, Japan) listed in S1 Table Finally, the heterozygous KO mice were mated with B6D2F1 and then maintained by sibling matings. The B6D2 KO mouse line is available from the Riken BioResource Center (Riken BRC, Tsukuba, Japan; #11261) and the Center for Animal Resources and Development, Kumamoto University (CARD, Kumamoto, Japan; #3019).

### Generation of Trim41 transgenic mice

The mouse *Trim41* cDNA (ENSMUST00000047145) tagged with PA and 1D4 was inserted under the control of the mouse *Clgn* promoter (Addgene #173686) [20,48]. The *Trim41* cDNA inserted plasmids are deposited at Riken BRC and Addgene (Addgene #186339). After linearization, the DNA construct (2.16 ng/μL; 0.54 ng/μL/kbp) was injected into the pronucleus of fertilized eggs. The injected eggs were transplanted into the oviduct ampulla of pseudopregnant mice (ICR; 10 embryos per ampulla). After 19 days, pups were delivered through Caesarean section and placed with foster mothers (ICR). For the rescue experiment, F0 Tg mice were mated with the *Trim41* KO mouse line. The mouse colony with the transgene and KO allele was maintained by sibling matings. The genotyping primers (GeneDesign) are listed in S1 Table The transgenic mouse line is available from Riken BRC (#11216) and CARD (#3019).

## Generation of ΔRING-Trim41 mice

*ΔRING-Trim41* mice were generated by microinjection described previously [49]. First, a gRNA solution was prepared by annealing two tracrRNAs (Sigma-Aldrich, St. Louis, MO, USA) and crRNA (Sigma-Aldrich). The target genomic sequences are listed in S1 Table Then, the gRNA solution and Cas9 nuclease solution (Thermo Fisher Scientific, Waltham, MA, USA) were mixed: 40 ng/µL gRNA each and 108 ng/µL Cas9 nucleases. The obtained complex was then microinjected into fertilized eggs (B6D2F1) using a programmable microinjector (FemtoJet 4i, Eppendorf, Hamburg, Germany). The microinjected eggs were then transplanted into the oviduct ampulla of pseudopregnant mice (ICR) on the following day. After 19 days, pups were delivered through Caesarean section and placed with foster mothers (ICR). To generate heterozygous mutant mice, F0 mice were mated with WT B6D2F1. Mouse colonies with the desired mutation were maintained by sibling matings. The genotyping primers (GeneDesign) are listed in S1 Table The mutant mouse line is available from Riken BRC (#11041) and CARD (#2948).

## Bacterial strains

*Escherichia coli* (*E. coli*) strain DH5α (Toyobo, Osaka, Japan) and BL21(de3) pLysS (C606003, ThermoFisher Scientific) were used for DNA cloning and protein expression, respectively. E. coli cells were grown in LB or 2×YT medium containing 100 mg/L ampicillin and were transformed or cloned using standard methods.

## Production of anti-TRIM41 antibody

A monoclonal antibody against TRIM41 was produced as previously described [16]. The DNA encoding mouse TRIM41 (residue 35–85 aa, NP_663352.2) was inserted into pGEX6p-1 (GE healthcare), and the expression vector was transformed into *E. coli* strain BL21 (de3) pLysS (C606003, Thermo Fisher Scientific). GST-TRIM41 was purified using Glutathione Sepharose 4B (GE Healthcare). The GST tag was removed by PreScission protease (27084301, GE Healthcare) and Glutathione Sepharose 4B affinity subtraction purification. The purified TRIM41 protein with a complete adjuvant was injected into female rats. After 17 days of injection, lymphocytes were collected from iliac lymph nodes for hybridomas generation [50,51]. The hybridomas were cloned by a limited dilution and screened by ELISA against recombinant TRIM41 and immunoblotting against testis lysate. A monoclonal antibody from hybridoma clone #33–16 was used in this study. As a side note, the #33–16 antibody also recognized ΔRING-TRIM41 missing 2–61 aa residues, showing that the epitope of #33–16 is in 59–85 aa residues (two or three mismatches might be acceptable).

## Genotype analysis

PCR was performed using KOD FX neo (KFX-201, TOYOBO). The primers (GeneDesign) for each gene are summarized in S1 Table PCR products were purified using a Wizard SV Gel and PCR Clean-Up System (Promega, Madison, WI, USA) kit for Sanger sequencing.

## Sequence comparison analysis

Amino acid sequences of TRIM41 were obtained from the NCBI Entrez Protein database. Clustal W2.1 was used for multiple sequence alignment [17].

## Immunoblotting

Proteins from testis were extracted using NP40 lysis buffer [50mM Tris-HCl (pH 7.5), 150 mM NaCl, 0.5% NP-40, 10% Glycerol]. Proteins were separated by SDS-PAGE under reducing

conditions and transferred to polyvinylidene fluoride (PVDF) membrane using the Trans Blot Turbo system (BioRad, Munich, Germany). After blocking with 10% skim milk (232100, Becton Dickinson, Cockeysville, MD, USA), the membrane was incubated with primary antibody overnight at 4°C, and then incubated with HRP-conjugated secondary antibody for 1 h at room temperature. Chemiluminescence was detected by ECL Prime Western Blotting Detection Reagents (RPN2232, GE Healthcare, Chicago, IL, USA) using the Image Quant LAS 4000 mini (GE Healthcare). The antibodies used in this study are listed in S2 Table

## Fertility analysis

To examine fertility, sexually mature male mice were caged with wild-type females (B6DF1) for at least three months. The vaginal plugs and pup numbers were recorded at approximately 10 AM to determine the number of copulations and litter size. Numerical data is available in S3 Table

## Morphological and histological analysis of testis and epididymis

To observe gross testis morphology and measure testicular weight, mice over 11 weeks of age were euthanized after measuring their body weight. Numerical data is available in S3 Table The whole testis was observed using BX50 and SZX7 (Olympus, Tokyo, Japan) microscopes. For histological analysis, testes were fixed in Bouin's fixative solution (16045–1, Polysciences, Warrington, PA, USA) at 4°C O/N, dehydrated in increasing ethanol concentrations and 100% xylene, embedded in paraffin, and sectioned (5 μm). The paraffin sections were hydrated with 100% Xylene and decreasing ethanol concentrations, treated with 1% periodic acid (26605–32, Nacalai Tesque, Kyoto, Japan) for 10 min, incubated with Schiff's reagent (193–08445, Wako) for 20 min, counterstained with Mayer's hematoxylin solution (131–09665, Wako) for 3 min, dehydrated in increasing ethanol concentrations, and finally mounted with Permount (SP15-100-1, Ferma, Tokyo, Japan). The sections were observed using a BX53 (Olympus) microscope. Seminiferous tubule stages were identified based on the morphological characteristics of the germ cell nuclei [52].

## Apoptosis detection in testicular sections

TdT-mediated dUTP nick end labeling (TUNEL) staining was carried out with In Situ Apoptosis Detection Kit (MK500, Takara Bio Inc., Shiga, Japan), according to the manufacturer's instruction. Briefly, testes were fixed with Bouin's fixative, embedded in paraffin, and sectioned (5 μm). After paraffin removal, the slides were boiled in citrate buffer (pH 6.0; 1:100; ab93678, abcam, Cambridge, UK) for 10 min and incubated in 3% $H_2O_2$ at room temperature for 5 min for endogenous peroxidase inactivation, followed by a labeling reaction with TdT enzyme and FITC-conjugated dUTP at 37°C for 1 h.

For chromogenic detection of apoptosis, the sections were incubated with HRP-conjugated anti-FITC antibody at 37°C for 30 min. The section was then incubated in ImmPACT DAB (SK-4105, Vector Laboratories, Burlingame, CA, USA) working solution, counterstained with Mayer's hematoxylin solution for 3 min, dehydrated in increasing ethanol concentrations, and finally mounted with Permount. The sections were observed using a BX53 (Olympus) microscope. Seminiferous tubule stages were identified based on the morphological characteristics of the germ cell nuclei [52]. Numerical data is available in S3 Table

## Sperm motility analysis

The sperm motility analysis was carried out as previously described [49]. In brief, spermatozoa extracted from cauda epididymis were incubated in TYH medium [53] at 37°C for 10 min.

Then, the incubated samples were diluted and analyzed using Hamilton Thorne's CEROS II sperm analysis system (software version 1.5.2; Hamilton Throne Biosciences, Beverly, MA).

## Immunostaining of surface chromosome spread

Spread nuclei from spermatocytes were prepared as previously described [15,16]. Seminiferous tubules were unraveled using forceps in ice-cold DMEM (11995065, Thermo Fisher Scientific) and incubated in 1 mg/mL collagenase (C5138, Sigma-Aldrich) in DMEM (20 mL) at 37°C for 15 min. After three washes with DMEM, the tubules were transferred to 20 mL trypsin/DNaseI medium [0.025 w/v% trypsin, 0.01 w/v% EDTA, 10U DNase in DMEM] and incubated at 37°C for 10 min. After adding 5 mL of heat-inactivated FCS and pipetting, the solution was filtered through a mesh (59 μm; N-N0270T, NBC Meshtec Inc., Tokyo, Japan) to remove the tubule debris. The collected testicular cells were resuspended in hypotonic solution [100 mM sucrose] and 10 μL of the suspension was dropped onto a glass slide with 100 μL of fixative solution [1% PFA, 0.1% (v/v) Triton X-100]. The slides were then air-dried and washed with PBS containing 0.4% Photo-Flo 200 (1464510, Kodak Alaris, NY, USA) or frozen for longer storage at -80°C.

The spread samples were blocked with 10% goat serum in PBS and then incubated with primary antibodies overnight at 4°C in blocking solution. After incubation with AlexaFlour 488/546-conjugated secondary antibody (1:200) at room temperature for 1 h, samples are counterstained with Hoechst 33342 and mounted with Immu-Mount. The samples were observed using a BX53 (Olympus) microscope. The antibodies used in this study are listed in S2 Table [54–56]. Numerical data is available in S3 Table.

As a side note, in DMC1/MLH1 staining, we detected secondary antibody cross-reaction against anti-SYCP3 antibodies in SYCP3-overloaded regions. Therefore, to minimize the effect of secondary antibody cross-reaction, the spread nuclei were first incubated only with an anti-DMC1/MLH1 antibody (overnight at 4°C) and a corresponding secondary antibody (at room temperature for 1 h). Then, the spread nuclei were incubated with the other primary antibodies (at room temperature for 1 h) and other secondary antibodies (at room temperature for 1 h).

## Immunostaining of metaphase/anaphase I cells

For cytological analysis of metaphase/anaphase I cells, seminiferous tubule squashes were performed as previously described [16,57]. Briefly, seminiferous tubules were incubated in fix/lysis solution [0.1% Triton X-100, 0.8% PFA in PBS] at room temperature for 5 min. Tubule bunches were then put on glass slides with 100 μL of fix/lysis solution, minced into ~ 3.0 mm segments with forceps, and arranged so that no tubule segment overlapped. After removing the excess amount of fix/lysis solution, a coverslip and pressure were applied to disperse cells, followed by flash freezing in liquid nitrogen for 15 sec, and removing the coverslip with forceps and a needle. For longer storage, the glass slides were kept at -80°C with the coverslip.

The slides were blocked and permeabilized in 10% goat serum and 0.1% Triton X-100 for 20 min in PBS, and incubated with primary antibody overnight at 4°C. After incubation with Alexa Flour 488/546-conjugated secondary antibody (1:200) at room temperature for 1 h, samples are counterstained with Hoechst 33342 (1:2000) and mounted with Immu-Mount. Z-stack images were taken using a BZ-X700 (Kyence, Osaka, Japan) microscope and stacked using ImageJ software. The antibodies used in this study are listed in S2 Table

## Giemsa staining of metaphase I chromosome spreads

For preparing metaphase chromosome spreads, seminiferous tubules were unraveled using forceps in ice-cold PBS and transferred to a 1.5-mL tube with 1 mL of accutase (12679–54,

Nacalai Tesque), followed by clipping the tubules, and a 5 min incubation at room temperature. After filtration with a mesh (59 μm; N-N0270T, NBC Meshtec inc.) and centrifugation, the cells were resuspended in 8 mL of hypotonic solution [1% sodium citrate] and incubated for 5 min at room temperature. The suspension was centrifuged and 7 mL of supernatant was aspirated. The cells were then resuspended in the remaining 1 mL of supernatant and 7 mL of Carnoy's Fixative [75% Methanol, 25% Acetic Acid] were added gradually while shaking. After 2 washes with Carnoy's Fixative, the cells were resuspended in ~ 0.5 mL of Carnoy's Fixative and dropped onto a wet glass slide. The slide was stained with Giemsa Stain Solution (079–04391, wako) and observed using a BX53 (Olympus) microscope.

## Multicolor fluorescence in situ hybridization (mFISH) followed by SYCP3 immunostaining

Testicular germ cells were suspended in 3 mL of hypotonic solution [0.075% potassium chrolide] and incubated for 20 min at room temperature. Then the cells were mixed with 1 mL of Carnoy's Fixative [75% Methanol, 25% Acetic Acid] to the suspension for fixing. After 5 mL of Carnoy's Fixative addition, the suspension was again mixed and centrifuged. After three washes with 5 mL of Carnoy's Fixative, the fixed cells were subjected to metaphase spreader Hanabi (ADSTEC, Funabashi, Japan) for spread sample preparation.

For multicolor FISH analysis, the spread samples were hybridized with 21XMouse (Meta-Systems, Altlussheim, Germany) according to the manufacturer's protocol. For denaturation of the nuclear DNA, the spread samples were incubated in 2×SSC for 30 min at 70˚C and then treated with 0.07 M NaOH for 1 min at room temperature. The denatured samples were washed with 0.1×SSC and 2×SSC for 1 min at 4˚C, respectively, and then dehydrated with 70%, 95%, and 100% ethanol. Multicolor FISH probes were denatured for 5 min at 75˚C and applied to the spread samples. After hybridization for 48 h at 37˚C in a humidified chamber, the spread samples were treated with 0.4× SSC for 2 min at 72˚C, washed in 2×SSC containing 0.05% Tween 20 for 30 sec at room temperature, and rinsed with distilled water.

Immunostaining of the spread nuclei with SYCP3 was performed as described above. After incubation of the spread samples with primary and secondary antibodies, the slides were covered by a coverslip with DAPI/Antifade (MetaSystemes, Altlussheim, Germany) and observed under a fluorescent microscope. Fluorescence images were captured using a high-sensitive digital camera (α7s, SONY, Tokyo, Japan), and the chromosome numbering of each synaptonemal complex was determined based on the fluorescence color.

## Immunoprecipitation

Proteins from seminiferous tubules were extracted using NP40 lysis buffer [50 mM Tris-HCl (pH 7.5), 150 mM NaCl, 0.5% NP-40, 10% Glycerol]. Protein lysates were mixed with 20 μL Protein G-conjugated magnetic beads (DB10009, Thermo Fisher Scientific) with 2.0 μg antibody. The immune complexes were incubated for 1 h at 4˚C and washed 3 times with NP40 lysis buffer. Co-immunoprecipitated products were then eluted by resuspension in 2x SDS sample buffer [125 mM Tris-HCl (pH 6.8), 10% 2-mercaptoethanol, 4% sodium dodecyl sulfate (SDS), 10% sucrose, 0.01% bromophenol blue] and 10 min incubation at 70˚C. The antibodies used in this study are listed in S2 Table

## Mass spectrometry and data analysis

Before MS analysis, half of the eluted amount was subjected to SDS-PAGE and silver staining (06865–81, Nacalai Tesque). The remaining half was subjected to mass spectrometry (MS) analysis. The proteins were reduced with 10 mM dithiothreitol (DTT), followed by alkylation

with 55 mM iodoacetamide, and digested by treatment with trypsin and purified with a C18 tip (GL-Science, Tokyo, Japan). The resultant peptides were subjected to nanocapillary reversed-phase LC-MS/MS analysis using a C18 column (25 cm × 75 um, 1.6 μm; IonOpticks, Victoria, Australia) on a nanoLC system (Bruker Daltoniks, Bremen, Germany) connected to a tims TOF Pro mass spectrometer (Bruker Daltoniks) and a modified nano-electrospray ion source (CaptiveSpray; Bruker Daltoniks). The mobile phase consisted of water containing 0.1% formic acid (solvent A) and acetonitrile containing 0.1% formic acid (solvent B). Linear gradient elution was carried out from 2% to 35% solvent B for 18 min at a flow rate of 400 nL/min. The ion spray voltage was set at 1.6 kV in the positive ion mode. Ions were collected in the trapped ion mobility spectrometry (TIMS) device over 100 ms and MS and MS/MS data were acquired over an *m/z* range of 100–1,700. During the collection of MS/MS data, the TIMS cycle was adjusted to 1.1 s and included 1 MS plus 10 parallel accumulation serial fragmentation (PASEF)-MS/MS scans, each containing on average 12 MS/MS spectra (>100 Hz), and nitrogen gas was used as the collision gas.

The resulting data were processed using DataAnalysis version 5.1 (Bruker Daltoniks), and proteins were identified using MASCOT version 2.6.2 (Matrix Science, London, UK) against the SwissProt database. Quantitative value (available in S4 Table) and fold exchange were calculated by Scaffold4 (Proteome Software, Portland, OR, USA) for MS/MS-based proteomic studies.

## Statistics and reproducibility

All error bars indicated standard deviation. The sample numbers were described in each legend or/and in the figure panel.

## Supporting information

**S1 Fig. *Trim41* is an evolutionarily conserved gene expressed highly in pachytene germ cells.** (A) The percentage of TF342569 annotated species based on the TreeFam database (Release 9; http://www.treefam.org/). Dark and light green show species with and without TF342569 annotation, respectively. (B) Protein sequence comparison of TRIM41 in big brown bat (XP_027991236.1), cat (XP_003980687.1), cattle (NP_001193094.1), chimpanzee (XP_016809993.1), dog (XP_038536985.1), human (NP_291027.3), mouse (NP_663352.2), pig (XP_020939058.1), and rat (NP_001128209.1).
(TIF)

**S2 Fig. Expression confirmation of *Clgn-Trim41 Tg*.** (A) RT-PCR using postnatal testis of *Trim41* KO-Tg male mice. (B) Immunoblotting analysis with an anti-PA antibody. GAPDH was used as a loading control. (C) Immunoblotting analysis with an anti-TRIM41 antibody raised against recombinant TRIM41 (35–85 amino acid residues). GAPDH was used as a loading control.
(TIF)

**S3 Fig. Histological analysis of juvenile *Trim41* KO male mice.** (A) Testis/bodyweight of *Trim41* Het and *Trim41* KO juvenile mice. PND8, PND12, PND16, and PND22 correspond to when the first wave of spermatogenesis reaches meiotic entry, zygotene stage, pachytene stage, and round spermatid occurrence, respectively. Testis/bodyweight: $0.89 \pm 0.07 \times 10^{-3}$ [PND8-Het], $0.99 \pm 0.06 \times 10^{-3}$ [PND12-Het], $1.72 \pm 0.08 \times 10^{-3}$ [PND16-Het], $2.58 \pm 0.21 \times 10^{-3}$ [PND22-Het], $0.86 \pm 0.05 \times 10^{-3}$ [PND8-KO], $1.00 \pm 0.09 \times 10^{-3}$ [PND12- KO], $1.41 \pm 0.15 \times 10^{-3}$ [PND16-KO], $1.80 \pm 0.21 \times 10^{-3}$ [PND22-KO]. Error bars indicate standard deviation. The numerical data are available in S3 Table (B) PAS staining of seminiferous tubules of juvenile

mice. PaSC: pachytene stage spermatocyte; RST: round spermatid. Red arrowheads indicate apoptotic germ cells.
(TIF)

**S4 Fig. Analysis of spermatozoa extracted from cauda epididymis.** (A) Cells extracted from cauda epididymis were suspended in PBS containing 1% glutaraldehyde. The cell suspension on a plastic dish was observed by an inverted microscope. (B)The fixed cell suspension was dropped onto a glass slide, and a coverslip and gentle pressure were applied. The slides were observed with an upright phase-contrast microscope. Black asterisks indicate spermatozoa with misshaped heads. (C) The percentage of spermatozoa with misshaped heads: 11/181, 6% [WT]; 56/5, 97% [KO]. (D) A violin plot of sperm tail length: 124 ± 8 μm [WT], 86 ± 23 μm [KO] (s.d.). The numerical data are available in S3 Table (E) Motility analysis of cauda-extracted cells. The first frame of the analyzed movie and trajectory of cells are shown. Red arrowheads in the KO image indicate cells with a flagella(-like) structure.
(TIF)

**S5 Fig. Immunostaining of zygotene–pachytene stage spermatocytes, related to Fig 3A–3E.** (A and B) Additional SYCP3/γH2AX immunostaining images of zygotene–pachytene stage spermatocytes from Het (A) and KO (B) male mice. The white dashed line shows nuclei of different cells. The white arrowhead indicates SYCP3 overloading. (C) KO pachytene stage spermatocytes stained with anti- SYCP3/γH2AX antibodies and DAPI. The white dashed line shows nuclei of different cells. White arrowheads indicate missing γH2AX/DNA signals. High magnified images are shown in red, green, and purple boxes.
(TIF)

**S6 Fig. Immunostaining of meiotic cohesins.** (A and B) SYCP3/REC8 immunostaining of spread nuclei of prophase spermatocytes collected from adult WT (A) and KO (B) male mice. High magnified images for the sex chromosomes are shown in red, green, and blue boxes. (C and D) SYCP3/RAD21L immunostaining of spread nuclei of prophase spermatocytes collected from adult WT (C) and KO (D) male mice. High magnified images for the sex chromosomes are shown in red, green, and blue boxes.
(TIF)

**S7 Fig. Phenotypical analysis of *Trim41* KO females.** (A) Gross morphology of female gonads from E15.5 embryos. (B) Immunostaining of female gonad sections. White arrowheads indicate intense SYCP3 signals. (C) Ovarian histology of *Trim41* Het and KO females. The ovaries were collected 10 h after intraperitoneal administration of hCG (58 h after PSMG administration).
(TIF)

**S8 Fig. Well-spread pachytene stage spermatocytes stained with anti-SYCP1/SYCP3 antibodies, related to Fig 5A and 5B.** (A) Representative well-spread pachytene stage spermatocytes are shown (The first 18 pachytene spermatocytes were shown). White arrowheads show the seamless connection of SYCP1/SYCP3 signals. White asterisks indicate tangled SYCP1/SYCP3 signals. (B) An example of a Het pachytene stage spermatocyte with a seamless connection of SYCP1/SYCP3 signals. (C) Quantification data for SYCP1/SYCP3 staining. Red boxes indicate the percentage of pachytene spermatocytes exhibiting seamless connection of SYCP1/SYCP3 signals (1/70 [Het]; 9/43 [KO]). Green boxes show the percentage exhibiting tangled SYCP1/SYCP3 signals (0/70 [Het]; 9/43 [KO]).
(TIF)

**S9 Fig. Immunostaining of HORMAD1.** (A and B) The spread nuclei of prophase spermatocytes collected from Het (A) and KO (B) male mice were immune-stained with anti-SYCP3 and -HORMAD1 antibodies. XYs indicate the sex chromosomes. White arrowheads show HOMRAD1 signals on autosomes in KO pachytene spermatocytes.
(TIF)

**S10 Fig. Immunostaining of prophase I spermatocytes collected from ΔRING mice, related to Fig 7A and 7B.** (A and B) The spread nuclei of prophase spermatocytes collected from adult *Trim41*$^{wt/ΔRING}$ (A) and *Trim41*$^{ΔRING/ΔRING}$. (B) male mice were immune-stained with anti-SYCP3 and -γH2AX antibodies. At least three male mice were analyzed. (C) The percentage of each meiotic prophase stage in immunostained spread nuclei samples in A and B. The numerical data is available in S3 Table.
(TIF)

**S11 Fig. Additional images for ΔRING-TRIM41 accumulation, related to Fig 7C–7F.** (A) SYCP3/HA immunostaining of zygotene–diplotene spermatocytes from *Trim41*$^{ΔRING/ΔRING}$ male mice. White and blue arrowheads indicate HA signals on and outside of SYCP3 axes, respectively. High magnified images for the sex chromosome were shown in red, green, and blue boxes. At least three male mice were analyzed. (B–E) SYCP3/TRIM41 immunostaining of pachytene spermatocytes from Het (B), KO (C), *Trim41*$^{wt/ΔRING}$ (D), and *Trim41*$^{ΔRING/ΔRING}$ (E) male mice. White arrowheads indicate HA signals on SYCP3 axes. High magnified images for the sex chromosome are shown in red, green, and blue boxes. At least three male mice were analyzed.
(TIF)

**S12 Fig. Immunostaining of DMC1.** (A and B) The spread nuclei of prophase spermatocytes collected from adult Het (A) and KO (B) male mice were immunostained with anti-SYCP3 and -DMC1 antibodies. White and yellow arrowheads indicate remaining DMC1 foci on overloaded SYCP3 and normal-looking SYCP3, respectively. (C) The number of DMC1 foci is shown as a scatterplot with average and error bars (s.d.). Heterozygous diplotene stage spermatocytes had no DMC foci. (D) The number of DMC foci in KO late-pachytene and diplotene stage spermatocytes is shown as a scatterplot. Plus and minus in SYCP3-OL indicate DMC1 foci on overloaded SYCP3 and normal-looking SYCP3, respectively. (E) The spread nuclei of prophase spermatocytes collected from adult *Trim41*$^{ΔRING/ΔRING}$ male mice were immunostained with anti-SYCP3, -HA, and -DMC1 antibodies. High magnified images are shown in red boxes.
(TIF)

**S13 Fig. Immunostaining of MLH.** (A and B) The spread nuclei of prophase spermatocytes collected from adult Het (F) and KO (G) male mice were immunostained with anti-SYCP3 and -MLH1 antibodies. Yellow arrowheads indicate MLH1 foci on the pseudo-autosomal regions (PARs). The white arrowhead shows MLH1 foci on seamless SYCP3 signals connecting the sex chromosomes and autosomes. The blue arrowhead shows MLH1 foci on SYCP3-overloaded regions of autosome. (C) The number of MLH1 foci is shown as a scatterplot with average and error bars (s.d.). (D) The number of MLH1 foci on SYCP3-overloaded regions. (E) The spread nuclei of prophase spermatocytes collected from adult *Trim41*$^{ΔRING/ΔRING}$ male mice were immune-stained with anti-SYCP3, -HA, and -MLH1 antibodies. High magnified images are shown in red boxes. Yellow and white arrowheads in the highly magnified images indicate MLH1 foci colocalized with HA signals and not, respectively.
(TIF)

**S14 Fig. Additional images of SYCP1 and BRCA1 immunostaining, related to Fig 7G and 7H.** (A) SYCP1/HA immunostaining of surface chromosome spreads from *Trim41*$^{\Delta RING/\Delta RING}$ testis. Magenta arrowheads indicate HA signals on the SYCP1 positive/negative boundary. High magnified images are shown in red, green, and blue boxes. At least three male mice were analyzed. (B) BRCA1/HA immunostaining of surface chromosome spreads from *Trim41*$^{\Delta RING/\Delta RING}$ testis. Red arrowheads indicate HA signals on the BRCA1 negative part of the X chromosome axes. High magnified images are shown in red boxes. At least three male mice were analyzed.
(TIF)

**S15 Fig. Immunostaining of ubiquitin.** (A and B) The spread nuclei of prophase spermatocytes collected from adult Het (A) and KO (B) male mice were immune-stained with anti-SYCP3 and -ubiquitin antibodies. XYs show the sex chromosomes.
(TIF)

**S16 Fig. Immunoprecipitation for interacting protein identification.** (A and B) Immunoprecipitation using an anti-TRIM41 antibody and WT testis lysate. TRIM41 immunoblotting (A) validated the immunoprecipitation capability of the anti-TRIM41 antibody and the success of immunoprecipitation experiments. Red and green arrowheads indicate TRIM41 and SYCP3, respectively. (C and D) Immunoprecipitation using an anti-TRIM41 antibody and *Trim41*$^{\Delta RING/\Delta RING}$ testis lysate. HA immunoblotting (C) validated the success of immunoprecipitation experiments. Black and green arrowheads indicate ΔRING-TRIM41 and SYCP3, respectively. (E and F) Immunoprecipitation using an anti-HA antibody and *Trim41*$^{\Delta RING/\Delta RING}$ testis lysate. HA immunoblotting (E) validated the success of immunoprecipitation experiments. Black and green arrowheads indicate HA-ΔRING-TRIM41 and SYCP3, respectively. (G and H) Mass analysis of co-IPed eluates using anti-TRIM41 (G) and anti-HA (H) antibodies. Proteins with 0 spectra in KO lysate are extracted and summarized in the table. The whole proteomics data are available in S4 Table (I) Immunoblotting of chromosome axis proteins using PND16 testis lysates. 4 Het and KO littermates were examined. PND16 testis was used to minimize the effect of cell population differences. (J) Immunoprecipitation of chromosome axis proteins, followed by immunoblotting using an anti-Ubiquitin antibody. PND16 testicular germ cells were lysed in NP40 lysis buffer [50 mM Tris-HCl (pH 7.5), 150 mM NaCl, 0.5% NP-40, 10% Glycerol] supplemented with a cocktail of protease inhibitor and a DUB inhibitor (20 μM PR619).
(TIF)

**S1 Table. Primers and gRNAs used in this study.**
(XLSX)

**S2 Table. Antibodies used in this study.**
(XLSX)

**S3 Table. Numerical data that underlies graphs.**
(XLSX)

**S4 Table. The quantitative value of mass spectrometry analysis.**
(XLSX)

## Acknowledgments

We wish to thank the members of both the Department of Experimental Genome Research, Animal Resource Center for Infectious Diseases, and NPO for Biotechnology Research and

Development for experimental assistance. We also thank Dr. Kusakabe and Dr. Tateno at the Department of Biological Sciences, Asahikawa Medical University for the technical consultation of mFISH experiments, and Dr. Julio M. Castaneda for the critical reading of the manuscript.

## Author Contributions

**Conceptualization:** Seiya Oura.

**Data curation:** Seiya Oura.

**Formal analysis:** Seiya Oura, Toshiaki Hino, Ayako Isotani.

**Funding acquisition:** Seiya Oura, Toshiaki Hino, Taichi Noda, Masahito Ikawa.

**Methodology:** Toshiaki Hino.

**Project administration:** Masahito Ikawa.

**Resources:** Seiya Oura, Takashi Satoh, Taichi Noda, Takayuki Koyano, Makoto Matsuyama, Shizuo Akira, Kei-ichiro Ishiguro.

**Software:** Seiya Oura, Toshiaki Hino.

**Supervision:** Masahito Ikawa.

**Visualization:** Seiya Oura, Toshiaki Hino.

**Writing – original draft:** Seiya Oura, Kei-ichiro Ishiguro, Masahito Ikawa.

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
