## [Editor Report · Decision Letter 0]

15 Feb 2022

Dear Dr Ikawa,

Thank you very much for submitting your Research Article entitled 'Trim41 is required to regulate the chromosome axis protein dynamics and meiosis in male mice.' to PLOS Genetics.

The manuscript, Review Commons history and revision plan were fully evaluated at the editorial level. Based on the reviews, we will not be able to accept this version of the manuscript, but we would be willing to review a much-revised version, as laid out in the revision plan. Upon resubmission, it is likely that the manuscript will undergo re-review. We cannot, of course, promise publication at that time.

Should you decide to revise the manuscript for further consideration here, your revisions should address the specific points made by each reviewer, according to the revision plan you submitted with the manuscript. We will also require a detailed list of your responses to the review comments and a description of the changes you have made in the manuscript.

If you decide to revise the manuscript for further consideration at PLOS Genetics, please aim to resubmit within the next 60 days, unless it will take extra time to address the concerns of the reviewers, in which case we would appreciate an expected resubmission date by email to plosgenetics@plos.org.

[LINK]

We are sorry that we cannot be more positive about your manuscript at this stage. Please do not hesitate to contact us if you have any concerns or questions.

Yours sincerely,

Paula E. Cohen

Associate Editor

PLOS Genetics

Gregory Barsh

Editor-in-Chief

PLOS Genetics

---

## [Decision Letter · Decision Letter 1]

5 Apr 2022

Dear Masa,

Thank you very much for submitting your Research Article entitled 'Trim41 is required to regulate chromosome axis protein dynamics and meiosis in male mice.' to PLOS Genetics.

The manuscript was fully evaluated at the editorial level and by independent peer reviewers. The reviewers appreciated the attention to an important topic but identified some minor concerns that we ask you address in a revised manuscript. These concerns can be addressed largely through manuscript editing, particularly explaining the generation of the mutant mouse lines, as requested by Reviewer 2. Additional changes are requested by all reviewers. We therefore ask you to modify the manuscript according to the review recommendations. Your revisions should address the specific points made by each reviewer.

[LINK]

Yours sincerely,

Paula E. Cohen

Associate Editor

PLOS Genetics

Gregory Barsh

Editor-in-Chief

PLOS Genetics

Reviewer's Responses to Questions

**Comments to the Authors:**

Reviewer #1: The authors corrected all my major comments on the previous version. The new data and edited text and figures are much improved.

Reviewer #2: In this manuscript, the authors analyse meiotic phenotypes of Trim41 knockout (KO) male mice and conclude that, surprisingly, this ubiquitin ligase E3 is essential for proper meiotic progression and fertility in male mice. The deletion of the RING domain phenocopy the KO mice suggesting that the E3 ubiquitin ligase activity is likely required for the meiotic functions. They show that spermatocytes of both mutant mice exhibit SYCP3 overloading in pachytene chromosomes, especially on the X chromosome. ΔRING-TRIM41 accumulated on the chromosome axes with overload SYCP3 which suggests a function on the chromosome axes. However they were unable to co-immunoprecipitate SYCP3 with TRIM41 or provide any direct molecular evidence of a direct interplay between them. The manuscript is merely descriptive and does not provide any mechanism to support their conclusions. Most if not all of the figures are wrongly numbered and ordered, which makes it difficult to read and review the paper. Plos Genetics is a journal of high quality and the articles that are submitted should be correctly edited to facilitate the work of the reviewers.

In the present MS the authors used three mouse models: one KO, one transgenic, and a ΔRING and performed comprehensive phenotypic analyses. However, the strategy to develop the KO is not explained at all and they do not show that the targeting strategy actually gives rise to a null allele due to the fact that the antigen (residue 35-85 aa) used for the production of anti-TRIM41 antibodies is deleted in the KO (Fig S2C). A crucial point of the MS would be if there is or not a truncated protein being expressed despite the presence of the Neo cassette. The targeting construct employed also appears to lack the RING domain. Thus, it seems quoite feasible that the two mutant models (KO and ΔRING) are indeed equivalent / redundant. If so, it would not be necessary to present the same experiments in the 2 models. The authors must explain the strategy used in both models (show the region eliminated in the KO and why they consider that a null allele is generated and not a hypomorphic allele or a protein without E3 ligase activity; include the ssODN, etc…) in order to clarify this relevant question.

In lane 224-225 the authors affirm that “The result showed the X chromosomes with more frequent SYCP3 overloading than autosomes (Fig. 4C; X chromosome: 37 out 43 cells; autosome: 12 out 43 cells)”.That is 86% showed the X chromosome with SYCP3 overloading vs 28% with an autosome. However in Fig 3D, 148/207 cells (72%) shows Sycp3 overloading on both sex chromosomes and autosomes. These results are confusing; in the experiment of Fig 3D it appears that in most cells (72%) both sex chromosomes and autosomes are affected but in Fig 4C only 28% of the cells show affected the autosomes. Moreover, most if not all of the pictures of the KO cells show Sycp3 overloading on both type of chromosomes. This is incosnsitent.

In relation to Fig 3F, do the authors find any “normal” metaphase/anaphase?? What percentage are abnormal?

In Figure 5E and table S3 the number of late pachytene cells analysed per mouse is too low to obtain a conclusion (6, 10, 5, 4, 12 and 9 cells). Authors should increase the number of cells analysed.

The numerical data of table S3 about the quantification of BRCA1 do not correspond to the text or to figure 5F. These data should be reviewed by the authors in order to clarify the conclusion reached.

Reviewer #3: In this manuscript, resubmitted from Review Commons, the authors present information on the male infertility phenotype of mice lacking TRIM41, a ubiquitin ligase E3 protein. They report on creation of KO of theTrim41 gene, originally to investigate its role in innate immunity. Surprisingly, however, the KO males were infertile, with arrest of spermatogenesis in the meiotic stage. The synaptonemal complex protein SYCP3 exhibited “overloading,” especially on the unsynapsed X chromosome. Mice lacking the RING domain of TRIM41 phenocopied the KO, providing evidence that the ubiquitin ligase E3 activity is essential for normal accumulation of SYCP3 on chromosome axes and especially for the unsynapsed X-chromosome dynamics. These results are important and noteworthy in that they provide new information about male meiotic X-chromosome dynamics and may lead to further defining the mechanisms that remodel and develop the chromosome axis. Overall, this report is a very interesting phenotype description that lays the foundation for additional investigations and will attract considerable attention in the fields of meiosis and reproductive genetics.

As one of the previous reviewers, I find that the authors have provided a well-reasoned response to the previous review critiques and the manuscript has been appropriately revised to mitigate previous concerns and avoid over-reaching conclusions. The quality of the figures and imaging of SYCP3 labeling have been improved. Importantly, the authors have clarified that while the phenotype informs us about processes requiring TRIM41, it does not reveal precise mechanisms by which TRIM41 acts. For instance, we do not know how its localization is regulated or its substrates, nor do we yet know which aspects of the meiotic infertility phenotype are direct or indirect effects of the deficiency in TRIM41. Nonetheless, these limitations do not diminish enthusiasm for this nice phenotype description, which is an essential step toward unraveling mechanisms.

Overall, descriptions of strategies and data are clearly presented, analyses are appropriately controlled, and the figures are of high quality. However, in the compiled manuscript PDF provided for review, there were multiple errors (probably at the level of Editorial Manager) in assembly and labeling of figures. The labels on the authors’ figures are correct and figures match the legends; the headers inserted by Editorial Manager at the top of each figure page are frequently incorrect, leading to assembly of figures in the wrong order. This confusion must be resolved.

**Have all data underlying the figures and results presented in the manuscript been provided?**

Reviewer #1: Yes

Reviewer #2: Yes

Reviewer #3: Yes

PLOS authors have the option to publish the peer review history of their article (what does this mean?). If published, this will include your full peer review and any attached files.

Reviewer #1: No

Reviewer #2: No

Reviewer #3: No

---

## [Decision Letter · Decision Letter 2]

6 May 2022

Dear Masa,

We are delighted to inform you that your manuscript entitled "Trim41 is required to regulate chromosome axis protein dynamics and meiosis in male mice." has been editorially accepted for publication in PLOS Genetics. Congratulations!

Yours sincerely,

Paula E. Cohen

Associate Editor

PLOS Genetics

Gregory Barsh

Editor-in-Chief

PLOS Genetics

Comments from the reviewers (if applicable):

Reviewer's Responses to Questions

**Comments to the Authors:**

Reviewer #2: The authors corrected all my major comments on the previous versión and the manuscript has been appropriately revised. The new data and figures are much improved.

**Have all data underlying the figures and results presented in the manuscript been provided?**

Reviewer #2: Yes

PLOS authors have the option to publish the peer review history of their article (what does this mean?). If published, this will include your full peer review and any attached files.

Reviewer #2: No

**Data Deposition**

http://datadryad.org/submit?journalID=pgenetics&manu=PGENETICS-D-22-00112R2

**Press Queries**

---

## [Editor Report · Acceptance letter]

25 May 2022

PGENETICS-D-22-00112R2 

*Trim41*is required to regulate chromosome axis protein dynamics and meiosis in male mice. 

Dear Dr Ikawa, 

We are pleased to inform you that your manuscript entitled "*Trim41*is required to regulate chromosome axis protein dynamics and meiosis in male mice." has been formally accepted for publication in PLOS Genetics! Your manuscript is now with our production department and you will be notified of the publication date in due course.

With kind regards,

Zsofia Freund

PLOS Genetics

On behalf of:
